# Objective identification of meteorological fronts and climatologies from ERA-Interim and ERA5

Philip G. Sansom[1,2] and Jennifer L. Catto[1]

[1]Faculty of Environment, Science and Economy, University of Exeter, North Park Road, Exeter, EX4 4QE, UK

[2]Met Office, FitzRoy Road, Exeter, EX1 3PB, UK

**Correspondence:** Jennifer L. Catto (j.catto@exeter.ac.uk)

**Abstract.** Meteorological fronts are important due to their associated surface impacts, including extreme precipitation and extreme winds. Objective identification of fronts is therefore of interest in both operational weather prediction and research settings. The aim of this study is to produce a front identification algorithm, based on earlier studies, that is portable and scalable to different resolution datasets. We have made a number of changes to an earlier objective front identification algorithm, applied these to reanalysis datasets, and present the improvements associated with these changes. First, we show that a change in the order of operations yields smoother fronts with fewer breaks. Next, we propose the selection of the front identification thresholds in terms of climatological quantiles of the threshold fields. This allows for comparison between datasets of differing resolutions. Finally, we include a number of numerical improvements in the implementation of the algorithm, and better handling of short fronts, which yield further benefits in smoothness and number of breaks. This updated version of the algorithm has been made fully portable and scalable to different datasets in order to enable future climatological studies of fronts and their impacts.

## 1 Introduction

Atmospheric fronts are of great importance for the day-to-day variability of weather in the mid-latitudes. They are associated with a large proportion of both total and extreme precipitation, as demonstrated by case studies (Browning, 2004), modelling (Browning, 1986; Sinclair and Keyser, 2015), and more recently, long-term climatologies (Berry et al., 2011b; Parfitt et al., 2017b; Schemm et al., 2017). They are also strongly linked to extreme wind events (Dowdy and Catto, 2017; Catto et al., 2019; Raveh-Rubin and Catto, 2019; Catto and Dowdy, 2021), and are key for air-sea interaction (Parfitt et al., 2017b). With a wealth of global gridded observationally-constrained and model-produced data, there is a desire to be able to objectively identify these frontal features in the gridded data. This avoids the huge time requirements of a manual analysis, and allows the features to be linked to high impact weather, such as extreme precipitation or winds (Catto et al., 2012; Catto and Pfahl, 2013; Dowdy

and Catto, 2017). The application of the methods to model data of historical and future climate also allows the models to be evaluated for their ability to capture the dynamical features and their connection to precipitation events (Leung et al., 2022), and to investigate the future of such features and how they may impact water resources and natural hazards (Catto et al., 2014).

A number of methods have been developed to perform such objective identification. Hewson (1998, referred to hereafter as H98) compiled a summary of methods used to identify frontal features in gridded data, and further developed the methods based on a thermal front parameter. Thomas and Schultz (2019) highlighted the three main factors required in identifying fronts with such a thermal front parameter: first, the thermal variable and vertical level to be considered, e.g., temperature, potential temperature, or equivalent (or wet-bulb) potential temperature at $850\,\mathrm{hPa}$; second, a function of the variable, e.g., the gradient, or some second or third derivative; and finally, some thresholds. They found that different thermal variables each had pros and cons, and could be selected depending on the purpose of the study. The study by Jenkner et al. (2010) used equivalent potential temperature and its second derivative to place the frontal lines. This results in the fronts lying in the centre of a frontal zone, rather than at the leading edge as a synoptic meteorologist would typically put them. Berry et al. (2011b) directly applied the methods of H98 to gridded data at $2.5° \times 2.5°$ resolution, placing fronts on the warm side of the strong temperature gradient. This also included the addition of a numerical line-joining algorithm, which is used to link the frontal points into line features.

Other methods have used dynamical information to identify fronts. Simmonds et al. (2012) used information solely on wind shifts. This method was found to work better in the Southern Hemisphere than the Northern Hemisphere by Schemm et al. (2015). A combination of this and the thermal method was used by Bitsa et al. (2021) to identify cold fronts in the Mediterranean, with the method tailored to suit the smaller spatial scale of fronts in this region. Parfitt et al. (2017b) used a combination of vorticity and temperature, requiring both a thermal gradient and a wind shift. While each method has its advantages and disadvantages, many of the methods typically identify many of the same features (Hope et al., 2014).

A major difficulty in applying objective front identification is the many datasets and differing resolutions. This is particularly an issue when using gradients of thermal properties, since the resolution of the data will have a large impact on these gradients. The thresholds used to define fronts need to be varied depending on the resolution. Recently, Soster and Parfitt (2022) investigated the sensitivity of results to the use of different datasets and found a large difference in front frequency between the datasets. Higher resolution datasets consistently show higher frequency of frontal points, with the differences reduced when re-gridded to a common grid. This was shown to lead to large differences between datasets in the proportion of precipitation attributed to fronts.

Despite the many methods of identifying fronts, and issues and uncertainties associated with each of them, the thermal front parameter method of H98 has been successful in identifying the key climatological features of front frequency and the link to other variables in a number of studies (e.g., Berry et al., 2011b, a; Catto et al., 2012; Catto and Pfahl, 2013; Catto et al., 2014; Dowdy and Catto, 2017). Those studies used either the European Centre for Medium-Range Weather Forecasts' (ECMWF) ERA-40 reanalysis (Uppala et al., 2005) at $2.5° \times 2.5°$ resolution, or later the ECMWF ERA-Interim (Dee et al., 2011) reanalysis at $0.75° \times 0.75°$ resolution. However, the code used in those studies was not easily portable due to being written in a mixture of the NCAR Command Language (NCL, 2011) and FORTRAN 77 (1978), and did not easily scale to the ECMWF ERA5 reanalysis at $0.25° \times 0.25°$ or other high resolution datasets. The aim of this study is to create a portable

implementation of the front identification method of H98, that is able to scale to contemporary high resolution (re-)analyses with horizontal grid-spacings of $0.25°$ or less. We demonstrate a quantile based method of tuning the thresholds. First the data used are described in Section 2. Section 3 gives a description of the thermal front parameter method, and the improvements over the previous implementation of the algorithm. In Section 4 we compare the front climatology using the new method with previous methods and different datasets. We finish in Section 5 with a discussion of the benefits and challenges associated with such objective identification methods.

## 2   Data

The updated front identification procedure is applied to the ECMWF ERA-Interim reanalysis (ECMWF Reanalysis - Interim, Dee et al., 2011). The data used here have a resolution of $0.75° \times 0.75°$ on a regular longitude-latitude grid. The 6-hourly instantaneous air temperature and specific humidity fields at the $850\,\mathrm{hPa}$ level were used to compute the wet-bulb potential temperature $\theta_W$, using the direct method of Davies-Jones (2008, Equation 3.8), in order to identify fronts. The 6-hourly eastward and northward wind components at $850\,\mathrm{hPa}$ were used to compute the front speed using Equation 13 of H98, allowing classification into cold, warm or quasi-stationary fronts. ERA-Interim was chosen over the more recent ERA5 reanalysis (ECMWF Reanalysis v5, Hersbach et al., 2020) for the primary analysis since the updated procedure is of greatest benefit in middle and low resolution models, and the resolution of ERA-Interim is equal to that of the highest resolution among standard CMIP6 GCMs. Our baseline for comparison is the global climatology of fronts in ERA-Interim at $0.75° \times 0.75°$ produced by Dowdy and Catto (2017) using the method of Berry et al. (2011b). We also present a high resolution climatology based on applying the updated front identification procedure to the ERA5 reanalysis using the same 6-hourly fields as ERA-Interim but with a grid spacing of $0.25° \times 0.25°$.

## 3   Methodology

Following H98 and Berry et al. (2011b), fronts are identified in the wet-bulb potential temperature field $\theta_W$ at $850\,\mathrm{hPa}$. As described in H98 (their Equation 5), and implemented in Berry et al. (2011b), fronts are located as the zero contour of:

$$\nabla \cdot \nabla |\nabla \theta_W| = 0 \quad \text{or} \quad \nabla^2 |\nabla \theta_W| = 0. \tag{1}$$

For a one-dimensional front (Type 1 front in H98), this is simply the third derivative of the wet-bulb potential temperature $\theta_W$ (see Figure 3 of H98 for an intuitive explanation). We will refer to Equation 1 as the thermal front locator (TFL). In practice, most atmospheric fronts are curved and not simple one-dimensional objects. H98 derived an alternative (their Equation 6) to Equation 1, based on the computation of "five-point mean axes", designed to mitigate the effects of frontal curvature on the computation of Equation 1, which can lead to noise and exaggerated frontal curvature. Although the alternative definition was preferred by H98, we keep the definition in Equation 1 primarily for compatibility with Berry et al. (2011b) and the numerous

studies which have utilised that implementation. However, the option to use the alternative definition may be included in a future version of the code documented by this study.

H98 defined two additional criteria that must be met in order for a zero contour of Equation 1 to be considered a front. First, the rate of change of $\theta_W$ across the front in the direction of cold air must exceed some threshold value $K_1$. This criterion was formalised in Equation 9 of H98 as:

$$\nabla |\nabla \theta_W| \cdot \frac{\nabla \theta_W}{|\nabla \theta_W|} < K_1, \quad \text{where} \quad K_1 \leq 0\,\mathrm{K\,m^{-2}}. \tag{2}$$

This is the thermal front parameter (TFP) defined by Renard and Clarke (1965). For a one-dimensional front, this criterion simply states that the second derivative of $\theta_W$ must be negative, placing the front on the warm side of the gradient. Second, the gradient of $\theta_W$ in the adjacent baroclinic zone (ABZ) must be greater than some threshold value $K_2$. This criterion was formalised in Equation 11 of H98 as:

$$|\nabla \theta_W|_{ABZ} > K_2, \quad \text{where} \quad K_2 \geq 0\,\mathrm{K\,m^{-1}}, \tag{3}$$

with

$$|\nabla \theta_W|_{ABZ} = |\nabla \theta_W| + m\chi |\nabla |\nabla \theta_W||,$$

where $m = 1/\sqrt{2}$ and $\chi$ is the grid length. For a one-dimensional front, this criterion simply states that the magnitude of the gradient of $\theta_W$ must be greater than $K_2$ a fraction $m$ of a grid length in the direction of greatest increase in the gradient of $\theta_W$, i.e., inside the ABZ. The value of $m = 1/\sqrt{2}$ was suggested by H98 and we found it to be effective at the resolution of ERA-Interim ($0.75°$) and ERA5 ($0.25°$).

Fronts are identified as warm, cold or quasi-stationary using the front speed defined by Equation 13 of H98, which is given here as:

$$\frac{\mathbf{V} \cdot \nabla |\nabla \theta_W|}{|\nabla |\nabla \theta_W||} = K_3, \tag{4}$$

where $\mathbf{V} = (u, v)$ is the vector wind field at $850\,\mathrm{hPa}$. Following Berry et al. (2011b), we adopt a threshold of $K_3 = 1.5\,\mathrm{m\,s^{-1}}$ such that front points are defined as belonging to warm fronts if they have speed exceeding $1.5\,\mathrm{m\,s^{-1}}$, and as belonging to cold fronts if they have speed less than $-1.5\,\mathrm{m\,s^{-1}}$. All other front points are defined as belonging to quasi-stationary fronts.

The automatic front identification method described by Equations 1–4 has been re-implemented in the R statistical computing language (R Core Team, 2021). The new implementation includes one key methodological change described in Section 3.1, as well as number of numerical updates compared to that of Berry et al. (2011b).

## 3.1 Methodological changes

The intention of this study was to create a portable and scalable implementation of the front identification method of H98 as implemented by Berry et al. (2011b), since that implementation has been successfully used in a number of other studies (e.g., Berry et al., 2011a; Catto and Pfahl, 2013; Dowdy and Catto, 2017). However, one key methodological change was implemented regarding the order of operations when locating front objects as lines. Berry et al. (2011b) take what we will call a "mask-then-join" approach, illustrated in Figure 1a. First they locate all those grid boxes that satisfy the TFP criterion in Equation 2 to form a mask (the ABZ gradient criterion in Equation 3 is not used). Zero points of the TFL in Equation 1 are located by an exhaustive search using linear interpolation between only those grid boxes included in the TFP mask defined Equation 2. Finally, a line-joining algorithm is used to join the zero points of the TFL in Equation 1 into lines representing fronts. Points are joined to their nearest neighbour if the euclidean distance calculated in degrees of longitude and latitude between two points is less than a specified threshold. This requires the repeated calculation of the distance between the current point and all remaining un-joined points, making the algorithm computationally expensive. Berry et al. (2011b) also apply a minimum front length criteria of $250\,\mathrm{km}$.

In contrast, H98 originally proposed a "contour-then-mask" approach, which we adopt here and illustrate in Figure 1b. We identify zero points in the complete TFL field defined by Equation 1 and join them into lines using a contouring algorithm, specifically the `contourLines()` function in R. Zero points are again located by linear interpolation, but only zero points located in adjacent grid boxes are considered for joining into lines, reducing the computational expense compared to an exhaustive search and avoiding the need for repeatedly calculating the distance between large numbers of points. We then interpolate the values of the fields defined by the TFP and ABZ criteria in Equations 2 and 3 respectively onto the points located by the contouring algorithm. Only points that meet the TFP and ABZ criteria defined by Equations 2 and 3 are retained, leaving a set of pre-joined line segments representing fronts.

The two approaches are compared in Figure 1. Zero points in Equation 1 usually occur between grid points. That means that adjacent grid boxes meeting the TFP criteria in Equation 2 are required in order to find zero points of the TFL using Equation 1 by the mask-then-join approach. At or below the $0.75° \times 0.75°$ resolution of ERA-Interim, the region that satisfies the TFP criterion in Equation 2 is often narrow, frequently only one grid box wide. Therefore, the mask-then-join approach frequently fails to locate front points. This behaviour can be seen in Figure 1a where no front points are identified between $44.25°$N and $45.75°$N since two zonally adjacent grid boxes would be required for successful interpolation of a zero point between two masked points, given the orientation of the front. This may result in some features not being identified at all or, more frequently, gaps in what should be continuous features. The line-joining algorithm used by Berry et al. (2011b) attempts to mitigate this by using a search radius larger than one grid length, but this is only partially effective. In Figure 1a, the search radius is effective in joining the southern-most located point, but fails to bridge the gap between $44.25°$N and $45.75°$N to the region between $46.5°$N and $48.0°$N where multiple adjacent grid points might once again enable the location of zero points. The number of points located in that northern region is then too small to meet the minimum front length criteria on their own. The contour-then-mask approach approach originally proposed by H98 and demonstrated in Figure 1b is able to successfully

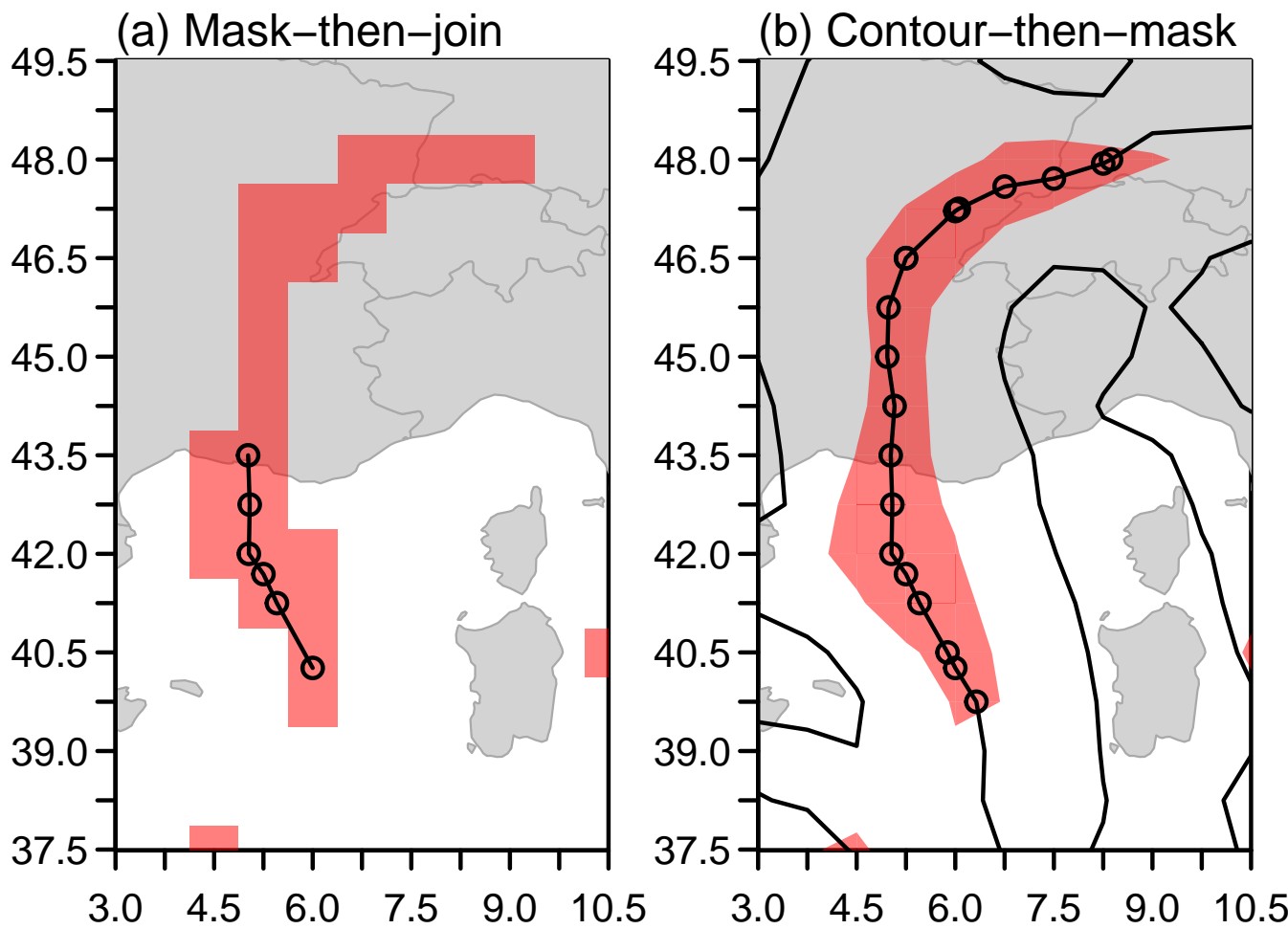

**Figure 1.** Front identification in ERA-Interim at 2001-01-01 00:00 UTC. (a) Using the mask-then-join approach, and (b) using the contour-then-mask approach. Black contours show $\nabla \cdot \nabla |\nabla \theta_W| = 0$ and red shading indicates regions where $\text{TFP} \leq -5 \times 10^{-11}\,\text{K}\,\text{m}^{-2}$ by masking in (a) and interpolation in (b). Circles indicate front points located by each algorithm.

identify the whole front as a single object. The masked region is shown for illustration only, in practice the masking variables are interpolated directly on to the potential front points located on the zero contour. Overall, the contour-then-mask approach results in more fronts and front points identified, and fewer breaks, as can be seen in the examples in Figures 3a and 3b, and the climatologies in Figures 4b and 4c. The expected decrease in the number of fronts due to there being fewer breaks is compensated by the number of new fronts located due to the increased sensitivity of the contour-then-mask approach to identifying potential front points. In some cases these new fronts were missed completely by the mask-then-join approach, in others they fail to meet the length criteria without additional points located by the contour-then-mask approach.

## 3.2 Choosing the thresholds and level of smoothing

Although automated methods offer the promise of objective feature identification, it is still usually necessary to set some key parameters subjectively. For the front identification method of H98 there are three parameters that require tuning: the amount of smoothing applied to the $\theta_W$ field, the TFP threshold $K_1$, and the gradient threshold $K_2$. Some studies have compared outputs with manual analyses by meteorologists to calibrate the parameters. While comparing to charts is a necessary check of an objective algorithm, calibrating in this way is difficult, time consuming and calibrates the algorithm to the subjective judgement of those meteorologists. Also, all three parameters depend on the resolution of the data. Therefore, the calibration must be repeated for each new dataset, or datasets brought to a common resolution for comparison. Instead, we offer some suggestions for objective calibration criteria.

We first address the smoothing problem, since the amount of smoothing applied to $\theta_W$ affects the choice of $K_1$ and $K_2$. The purpose of smoothing is to remove local minima and maxima that might break up otherwise continuous features. We particularly wish to avoid local extrema in the TFL field defined by Equation 1, which will appear as short closed contours of TFL $= 0\,\mathrm{K\,m^{-3}}$. Therefore, it makes sense to examine the effect of smoothing on the average length of the contours. The noise in the TFL field will in part be due to the choice to use Equation 1, as implemented by Berry et al. (2011b), to define the location of the fronts, rather than the method preferred by H98 (their Equation 6) designed to quell the amplification of frontal curvature. Previous studies applying the method of Berry et al. (2011b) to ERA-Interim used $n = 2$ passes of a simple five-point average to smooth the $\theta_W$ field. In testing on ERA-Interim data, it was found that the average length of the contours of TFL $= 0\,\mathrm{K\,m^{-3}}$ initially increases rapidly with the number of passes of the five-point smoother, but after 6–10 passes, the effect of further smoothing diminishes (see Figure 1 of Supplementary Material). Therefore, we settled on $n = 8$ passes of a five-point smoother.

Equation 1 was retained for its simplicity, and compatibility with Berry et al. (2011b) and subsequent studies. However, the alternative method preferred by H98 may be made available as an option in future versions of the code associated with this study. Jenkner et al. (2010) classify all closed contours in the front locating field encircling an area smaller than a given threshold as being associated with (potential) local, rather than synoptic fronts. Such a criterion introduces additional subjectivity, but would effectively reduce the noise when identifying synoptic fronts, possibly allowing less smoothing to be used, and further distinguish fronts associated with orography and other local features. The issue of noise and surface driven gradients was also discussed in Hewson (2001).

It is common to define weather phenomena as events exceeding some percentile of the climatological distribution. Therefore, we propose a quantile based approach to setting the thresholds $K_1$ and $K_2$. The advantage of setting thresholds in terms of climatological quantiles is that the thresholds should be comparable between datasets of differing resolution, while the actual values can differ quite widely. For example, Berry et al. (2011b) used a threshold for $K_1$ of $-8\times10^{-12}\,\mathrm{K\,m^{-2}}$ at 2.5° resolution in ERA-40, compared to the threshold of $-5\times10^{-11}\,\mathrm{K\,m^{-2}}$ at 0.75° resolution used in ERA-Interim by Dowdy and Catto (2017). In order to compute quantiles, we require climatologies of the TFP and the magnitude of the gradient, obtained by evaluating Equations 2 and 3, respectively, over an extended time period for the region of interest. The time period considered

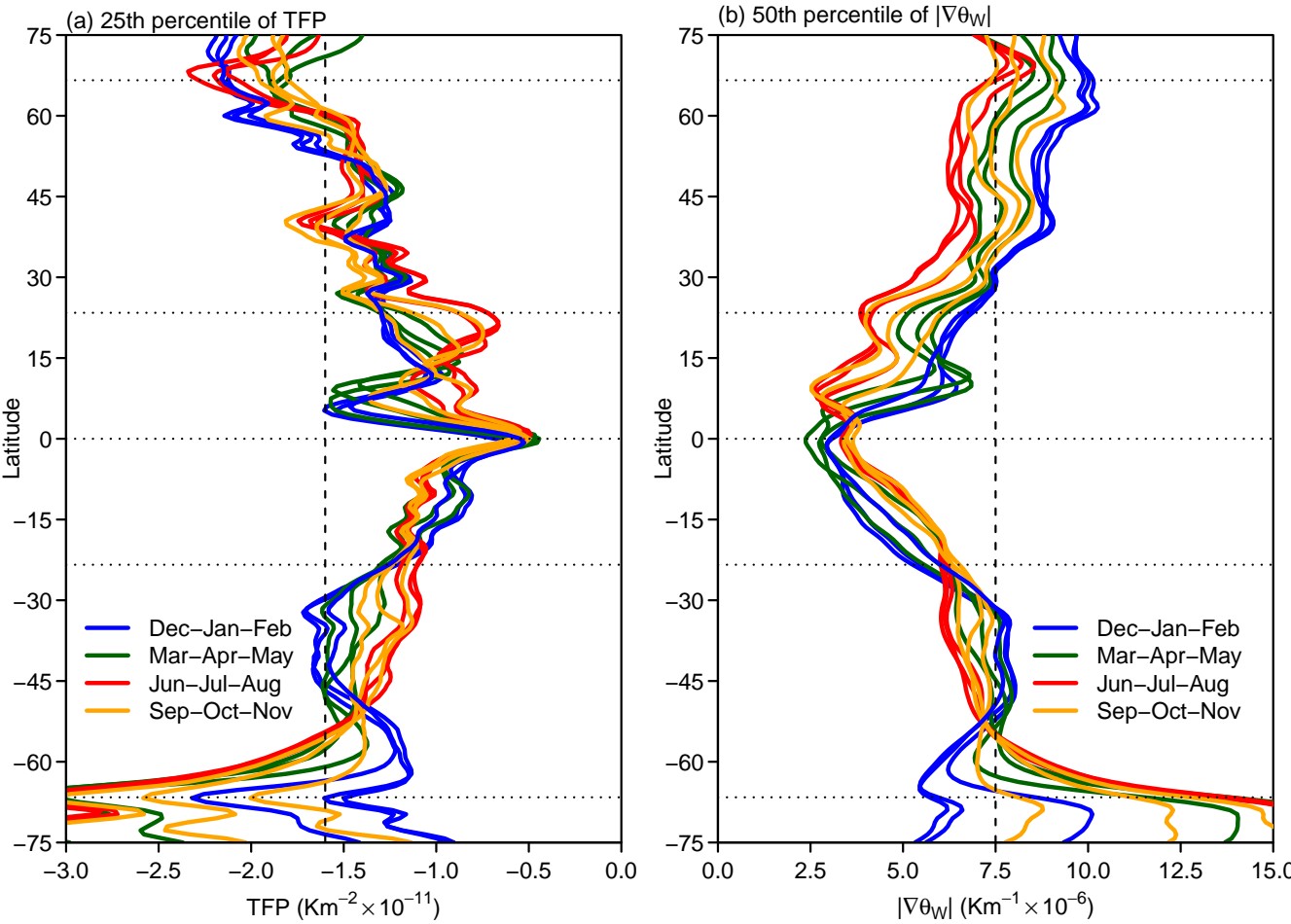

**Figure 2.** Choosing the thresholds $K_1$ and $K_2$ from ERA-Interim data (1979–2018) with $n = 8$ smoothing passes. (a) The 25th percentile of the TFP, and (b) the 50th percentile of $|\nabla\theta_W|$ by latitude and month of the year. Each coloured line represents a different month: blue for December, January and February, yellow for March, April and May, red for June, July and August, and orange for September, October and November. Horizontal dotted lines represent the major circles of latitude (66.6°N, 23.4°N, 23.4°S, 66.6°S). Vertical dashed lines indicate the thresholds chosen in the text: $-1.6 \times 10^{-11}\,\mathrm{K\,m^{-2}}$ in (a); and $7.5 \times 10^{-6}\,\mathrm{K\,m^{-1}}$ in (b).

was 1979-2018 in ERA-Interim. Since most fronts occur in the extra-tropical regions, we will focus our attention there. We seek quantiles of the TFP and the magnitude of the gradient that produce continuous fronts in good agreement with published charts for the North Atlantic and Europe, focusing on January and July 2020. Combinations of quantiles of both the TFP and the magnitude of the gradient were systematically compared (see Supplementary Material for examples). We set the first

190    threshold $K_1$ to the 25th percentile (0.25 quantile) of the climatological distribution of the TFP (See Supplementary Figure 2). In the Northern Hemisphere extra-tropics this is around $-1.6 \times 10^{-11}\,\mathrm{K\,m^{-2}}$. We set the second threshold $K_2$ equal to the 50th percentile (0.50 quantile) of the climatological distribution of the magnitude of the gradient of $\theta_W$ (see Supplementary Figure 3). In the Northern Hemisphere extra-tropics (23.4°N–66.6°N) this is around $7.5 \times 10^{-6}\,\mathrm{K\,m^{-1}}$. These choices are subjective,

and an operational meteorologist might make other choices. However, in the absence of strong physical reasoning, these quantiles have a simple symmetry, i.e., each is approximately the 50th percentile of the allowed range (since $K_1 < 0\,\mathrm{K\,m^{-2}}$ and globally the 50th percentile of TFP is approximately $0\,\mathrm{K\,m^{-2}}$), and produce continuous fronts in good agreement with published charts (Supplementary Figure 4).

Figure 2 illustrates the monthly and latitudinal climatological variation in the chosen quantiles of TFP and the magnitude of the gradient of $\theta_W$. The distributions of both TFP and the magnitude of the gradient are very different in the tropics compared to the extra-tropics. The value of TFP chosen for $K_1$ is biased towards the upper latitudes of the Northern Hemisphere extra-tropics where fronts are frequently observed and associated with extra-tropical cyclones. The chosen value of the magnitude of gradient lies in the middle of the seasonal variation in the extra-tropics, which is fairly constant between around 35°N–65°N and 30°S–50°S, with greater spread in the Northern Hemisphere. The chosen values are broadly representative of the quantiles across the seasons in both the northern and Southern Hemisphere extra-tropics. Given the relative insensitivity to reasonable values of the $K_1$ and $K_2$ shown in the Appendix, the chosen values should be representative across the seasons and both hemispheres for both criteria.

For comparison, previous studies applying the method of Berry et al. (2011b) to ERA-Interim used a threshold of $K_1 = -5 \times 10^{-11}\,\mathrm{K\,m^{-2}}$ after $n = 2$ smoothing passes (e.g., Dowdy and Catto, 2017; Catto and Raveh-Rubin, 2019; Raveh-Rubin and Catto, 2019; Catto and Dowdy, 2021). The ABZ gradient threshold $K_2$ in Equation 3 was not implemented by Berry et al. (2011b), equivalent to setting $K_2 = 0\,\mathrm{K\,m^{-1}}$ since $|\nabla\theta_W| \geq 0$ by definition. Our threshold $K_1$ is higher primarily due to the additional smoothing, but the exclusion of the second threshold $K_2$ may have caused Berry et al. (2011b) to choose a lower threshold for $K_1$ in order to remove unwanted features that could more effectively have been eliminated by implementing the second threshold $K_2$.

### 3.3 Comparing fronts from different datasets

When comparing analyses from different weather and climate datasets, the most common approach is to interpolate all the datasets to a common resolution, usually the lowest resolution among them. For some features such as fronts that are more easily identified in higher resolution data, this can be limiting. The objective calibration criteria described in Section 3.2 provide one route by which fronts could be identified at the native resolution of each dataset and then compared. The quantile based criteria will identify the same fraction of grid boxes potentially containing front points for any reasonable resolution and number of smoothing passes. However, computing the required climatologies is time consuming. An alternative is to keep the thresholds $K_1$ and $K_2$ constant, and adjust the number of smoothing passes such that the climatological distributions of the TFP and the magnitude of the gradient are similar between datasets. Specifically, the quantiles used to set the thresholds should be similar. In testing, it was found that matching the threshold quantile of the TFP field provided a more consistent comparison than that of the gradient field. It is sufficient to compare the quantiles for only a small subset of the data, provided the same subset is used for each dataset, avoiding the need to compute a long climatology in order to determine. In testing, various lengths and spatial extents of training data were considered for comparing ERA-Interim and ERA5, from one month, to 30 years, for the Northern Hemisphere extra-tropics, Southern Hemisphere extra-tropics, or the whole globe. One month

of data was found to be sufficient to consistently determine an appropriate number of smoothing passes. The procedure is not sensitive to either the month of the year or the spatial extent, among those considered. In practice, we used January 2001 for the Northern Hemisphere extra-tropics, consistent with the examples in Figures 1 and 3.

### 3.4 Numerical updates

Berry et al. (2011b) used repeated applications of a simple central finite difference approximation to the first derivative to evaluate all the derivatives in Equations 1–4 at each grid box. The simple approximation uses one grid box on either side of the box in question to approximate the first derivative to second order accuracy. The zonal and meridional derivatives are evaluated separately using one box to the left and right, or above and below, respectively. However, repeated applications of the approximation to the first derivative degrades the accuracy for higher derivatives. In contrast, we use an explicit central finite difference approximation to the second derivatives required to evaluate $\nabla^2 \theta_W = \frac{\partial^2 \theta_W}{\partial x^2} + \frac{\partial^2 \theta_W}{\partial y^2}$ when computing the TFL in Equation 1, avoiding the need for repeated applications of the first derivative, and maintaining second order accuracy. The zonal and meridional terms, $\frac{\partial^2 \theta_W}{\partial x^2}$ and $\frac{\partial^2 \theta_W}{\partial y^2}$ respectively, are evaluated separately. The computation of both the first and second derivatives was also updated to maintain second order accuracy at the edges of the domain using forward and backward differences. The increased accuracy at the edges has no additional computational cost, and the improved approximation to the second derivative is actually more efficient than repeated applications of the first derivative.

Other numerical differences include updates to the computation of relative humidity and wet-bulb potential temperature ($\theta_W$). Relative humidity is required to compute wet-bulb potential temperature. If only specific rather than relative humidity data are available, then relative humidity can be computed from temperature, specific humidity and pressure (which is constant at $850\,\mathrm{hPa}$). The implementation by Berry et al. (2011b) used the table based approach built into the NCAR Command Language to compute relative humidity. The new implementation uses the mixed-phase parametrization of relative humidity from the ECMWF Integrated Forecasting System (ECMWF, Section 7.4.2). In the new implementation, the wet-bulb potential temperature ($\theta_W$) is computed using the direct method of Davies-Jones (2008, Equation 3.8), rather than the iterative method implemented by Berry et al. (2011b). The final numerical difference between the two implementations is how short fronts are handled. In the original application, Berry et al. (2011b) reject any fronts less than three points long. In later applications this was updated to a great-circle distance based criterion where fronts whose end points are less than $250\,\mathrm{km}$ apart are rejected. In our implementation, we sum the great-circle distance between all adjacent points in each front and reject fronts whose total length is less than $250\,\mathrm{km}$. The length threshold of $250\,\mathrm{km}$ is a subjective choice that has been retained for approximate compatibility with previous studies.

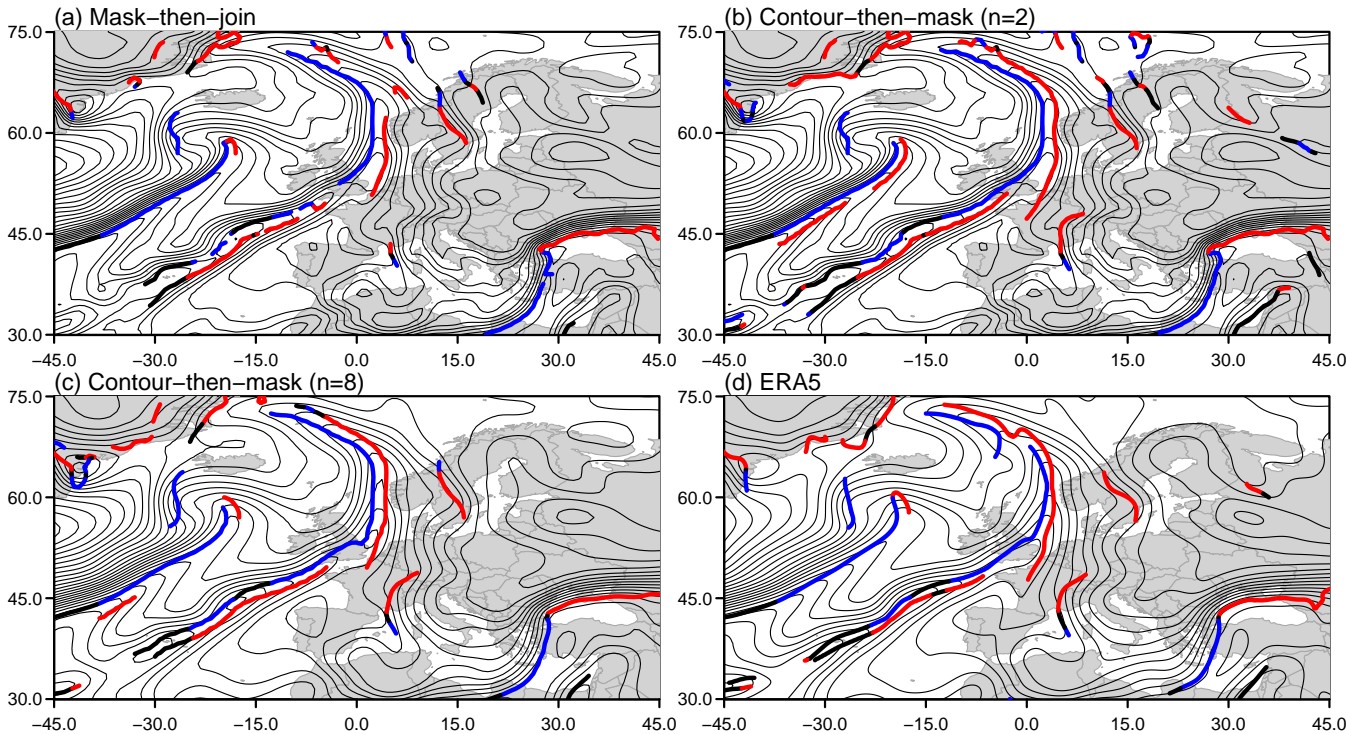

**Figure 3.** Comparison of methods in ERA-Interim at 2001-01-01 00:00 UTC. (a) mask-then-join with $n = 2$, $K_1 = -5 \times 10^{-11} \, \mathrm{K \, m^{-2}}$ and $K_2 = 0 \, \mathrm{K \, m^{-1}}$, (b) contour-then-mask with $n = 2$, $K_1 = -5 \times 10^{-11} \, \mathrm{K \, m^{-2}}$ and $K_2 = 0 \, \mathrm{K \, m^{-1}}$, (c) contour-then-mask with $n = 8$, $K_1 = -1.6 \times 10^{-11} \, \mathrm{K \, m^{-2}}$ and $K_2 = 7.5 \times 10^{-6} \, \mathrm{K \, m^{-1}}$, and (d) in ERA5 using contour-then-mask with $n = 96$, $K_1 = -1.6 \times 10^{-11} \, \mathrm{K \, m^{-2}}$ and $K_2 = 7.5 \times 10^{-6} \, \mathrm{K \, m^{-1}}$. Thin black lines indicate contours of wet-bulb potential temperature $\theta_W$. Thick blue lines indicate cold fronts, thick red lines indicate warm fronts and thick black lines indicate quasi-stationary fronts. All fronts were classified using a threshold of $K_3 = 1.5 \, \mathrm{m \, s^{-1}}$

## 4 Results

### 4.1 Comparison with previous implementations

Figure 3 illustrates the difference between the mask-then-join and contour-then-mask methods, and the effect of the updated parameter choices (i.e., $n$, $K_1$, and $K_2$) in ERA-Interim at 2001-01-01 00:00 UTC . The mask-then-join approach using the original parameters (Figure 3a) is clearly identifying fronts, but they are fractured with frequent gaps. The contour-then-mask (Figure 3b) results in much smoother front features with fewer gaps, and more fronts identified. Figure 3c shows the results of the updated parameters with more smoothing cycles and stronger thresholds. Figure 3d shows the fronts identified in ERA5, and will be discussed further in Section 4.3. Compared to the original parameters, the front features are smoother, with fewer breaks and many spurious local fronts have been removed. One feature that can be seen is a warm front running parallel to the (predominantly) cold front extending from the Azores across the south of the UK. Such features were noted by H98, and are associated with a warm conveyor belt running adjacent to the front. Hewson and Titley (2010) use a third masking criteria

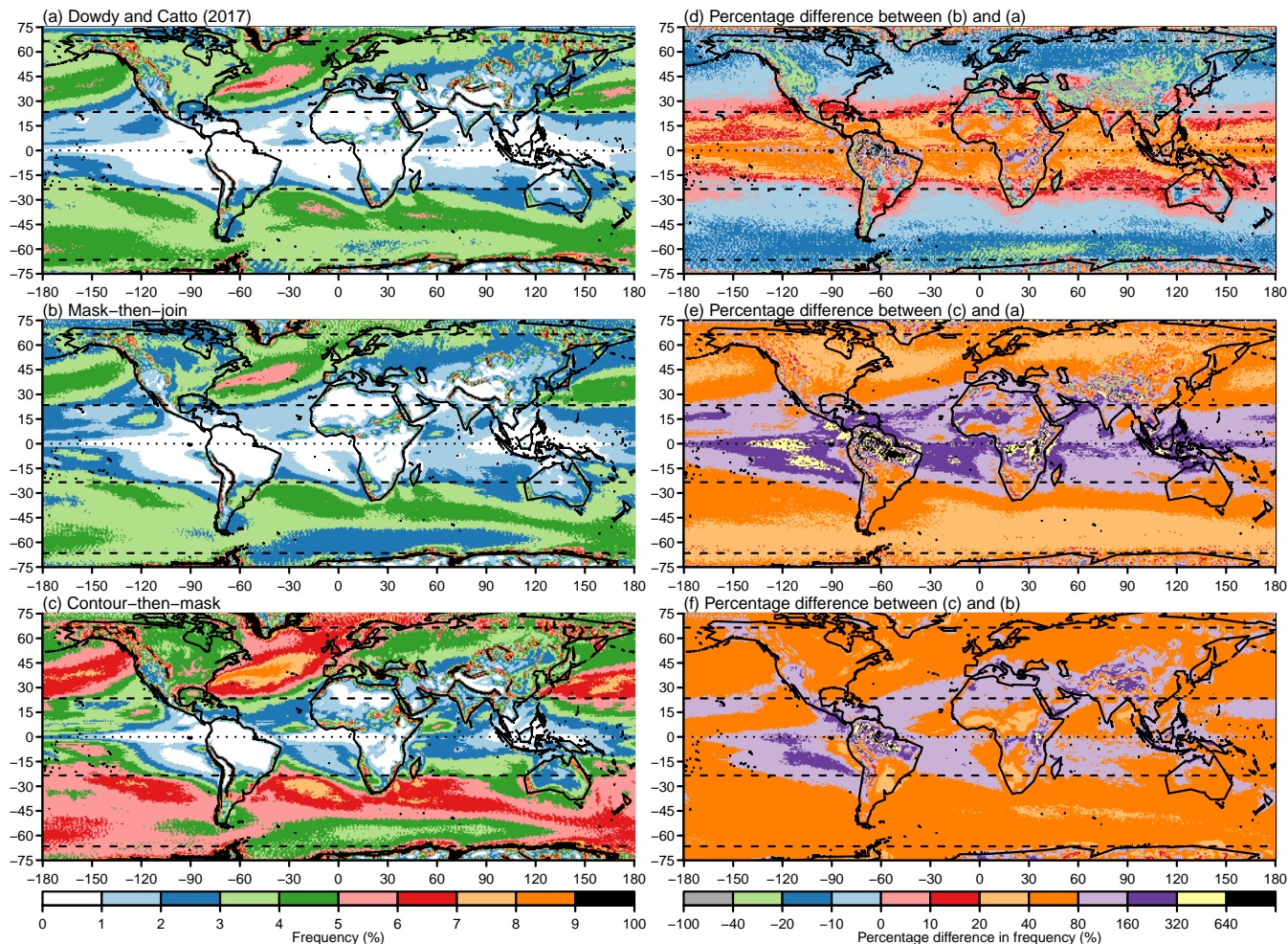

**Figure 4.** Comparison of global climatologies of front frequency (% of 6-hourly frames from 1979–2018). (a) Dowdy & Catto (2017), (b) updated implementation using the mask-then-join approach (c) updated implementation using the contour-then-mask approach, (d) percentage difference between mask-then-join and Dowdy & Catto (2017) ((b-a)/a), (e) percentage difference between contour-then-mask and Dowdy & Catto (2017) ((c-a)/a), and (f) percentage difference between contour-then-mask and mask-then-join ((c-b)/b). All climatologies were computed with $n = 2$ smoothing cycles, $K_1 = -5 \times 10^{-11}\,\mathrm{K\,m^{-2}}$ and $K_2 = 0\,\mathrm{K\,m^{-1}}$

based on potential temperature rather than wet-bulb potential temperature that may be implemented in a future version of the code documented in this study.

Figure 4 compares the front frequency climatologies from three different implementations of the H98 algorithm applied to ERA-Interim with identical parameters (i.e., the same number of smoothing passes, and thresholds $K_1$ and $K_2$): the implementation of Berry et al. (2011b) used by Dowdy and Catto (2017) (Figure 4a); a version incorporating our numerical updates but using the original mask-then-join approach (Figure 4b); and our final version using the contour-then-mask approach (Figure 4c). Figure 4d shows that our numerical updates result in slightly lower numbers of fronts identified in across most of the Northern and Southern Hemisphere extra-tropics, and slightly higher numbers of fronts identified in the tropics. Figures 4e and

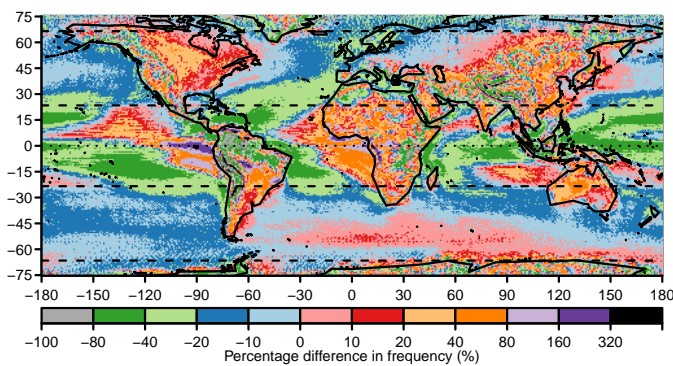

**Figure 5.** Updated parameters. Percentage difference between ERA-Interim climatology of front frequency computed using updated parameters $n = 8$, $K_1 = -1.6 \times 10^{-11} \, \mathrm{K \, m^{-2}}$ and $K_2 = 7.5 \times 10^{-6} \, \mathrm{K \, m^{-1}}$, and the original parameters $n = 2$, $K_1 = -5 \times 10^{-11} \, \mathrm{K \, m^{-2}}$ and $K_2 = 0 \, \mathrm{K \, m^{-1}}$.

4f compare our final version with the implementation of Berry et al. (2011b) and the version incorporating only the numerical updates. The numerical updates produce relatively modest differences in the number of fronts identified (Figure 4d). Small decreases are seen in the extra-tropics, and larger increases in the tropics. The move from the mask-then-join to the contour-then-mask approach has a greater effect in the extra-tropics (Figure 4f). In the Northern and Southern hemisphere storm tracks, the number of fronts identified increases by between 40 and $80 \, \%$. The increases are a combination of increases in the length of previously identified fronts, and the addition of fronts that were not previously identified, as demonstrated in Figures 1 and 3. The greatest relative increase in the number of fronts identified is seen in the tropics. Comparing Figures 4e and 4f shows that this is due to a combination of the numerical updates and the move to the contour-then-mask approach. The relative increase in the tropics is very large due to the scarcity of fronts in the tropics making even a small increase seem large. The absolute number of fronts detected in the tropics remains small compared to the extra-tropics (4c).

While changing the implementation of the front identification leads to an increase in the number of fronts identified, as shown in Figure 4, the next aspect of the updated method is a change to the parameters used. Figure 5 compares the climatology of front frequency of our final version with updated parameters (i.e., smoothing passes, and thresholds $K_1$ and $K_2$; shown in Figure 6a) applied to ERA-Interim against the implementation by Berry et al. (2011b) with the original parameters. The updated parameters result in slightly fewer fronts identified in almost all regions, due to the increased smoothing, making the climatology more similar to earlier estimates, but with the smoother individual fronts given by the contour-then-mask method. The greatest decreases are seen on the edges of the tropics, adjacent to regions with high front activity. This pattern is to be expected due to the rapid drop-off in the climatological quantile values of the masking parameters in the tropics in Figure 2.

## 4.2 Front climatology from ERA-Interim

Figure 6 shows the climatology of front frequency of our final version with updated parameters applied to ERA-Interim, including the breakdown into cold, warm and quasi-stationary fronts. Figure 6a allows for a direct comparison with the climatologies in Figure 4, showing that while the updated parameters reduce the number of fronts identified compared to the updated nu-

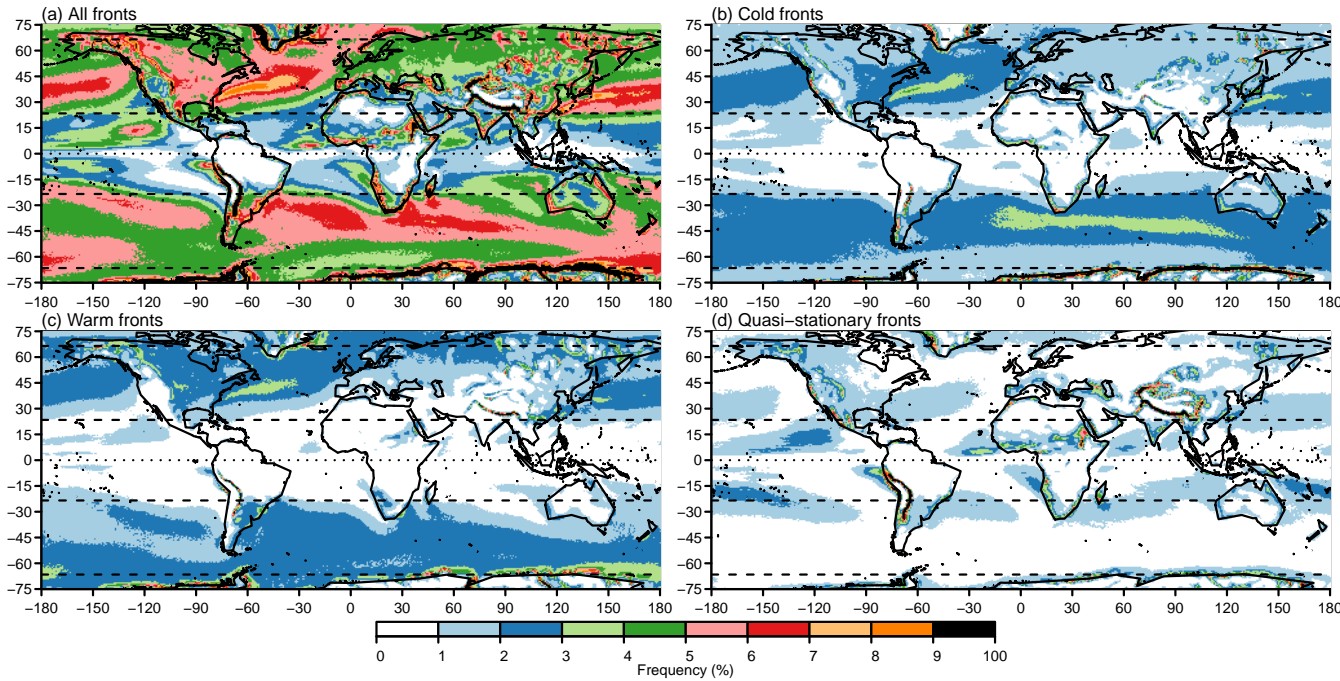

**Figure 6.** Updated global climatologies of front frequency as a percentage of times. (a) All fronts, (b) cold fronts, (c) warm fronts, (d) quasi-stationary fronts. All climatologies were computed with $n = 8$ smoothing cycles, $K_1 = -1.6 \times 10^{-11}\,\mathrm{K\,m^{-2}}$ and $K_2 = 7.5 \times 10^{-6}\,\mathrm{K\,m^{-1}}$ and $K_3 = 1.5\,\mathrm{m\,s^{-1}}$.

merical implementation only, overall more fronts are still identified in almost all regions than in earlier versions. Figure 6b and 6c show that cold and warm fronts occur with similar frequencies in most extra-tropical regions, as previously shown in Berry et al. (2011b). Figure 6d shows that quasi-stationary fronts occur most often where winds are weaker, particularly in the

300    horse latitudes close to $30°$N and $30°$S, the inter-tropical convergence zone (ITCZ) close to the Equator, and adjacent to high orography, as expected.

Figure 7 breaks the classification of fronts down still further, showing cold and warm fronts by season. Unsurprisingly, cold fronts in the Northern Hemisphere are most common at the beginning of the storm track regions of both the Atlantic and Pacific oceans in northern winter (DJF, Figure 7a). In contrast, warm fronts in northern summer (JJA, Figure 7g) tend to outnumber

305    cold fronts (Figure 7c). In agreement with Berry et al. (2011b), the seasonal distribution of fronts in the Southern Hemisphere is much more stable. Cold fronts are slightly more common though less widely distributed in the Southern Hemisphere during southern summer (DJF, Figure 7a) than in southern winter (JJA, Figure 7c) consistent with Berry et al. (2011b) and the climatology of frontogenesis by Satyamurty and de Mattos (1989). The large numbers of warm fronts near Antarctica (JJA, Figure 7f,g,h) are likely related to the strong temperature gradients between the sea surface and sea ice.

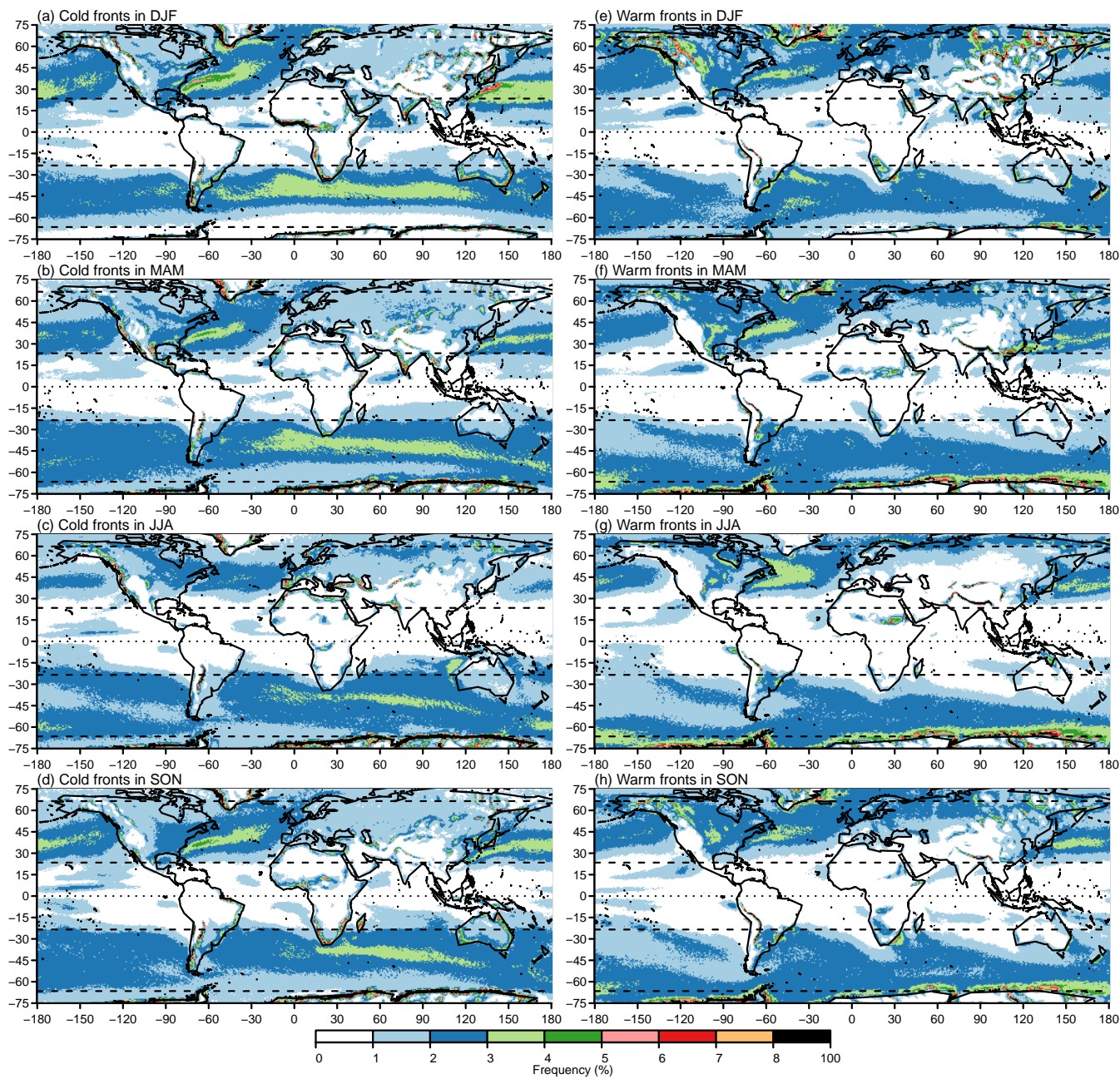

**Figure 7.** Updated seasonal climatologies of front frequency. Cold fronts (a-d), and warm fronts (e-h) for (a, e) DJF, (b, f) MAM, (c, g) JJA, (d, h) SON. All climatologies were computed with $n = 8$ smoothing cycles, $K_1 = -1.6 \times 10^{-11}\,\mathrm{K\,m^{-2}}$ and $K_2 = 7.5 \times 10^{-6}\,\mathrm{K\,m^{-1}}$ and $K_3 = 1.5\,\mathrm{m\,s^{-1}}$.

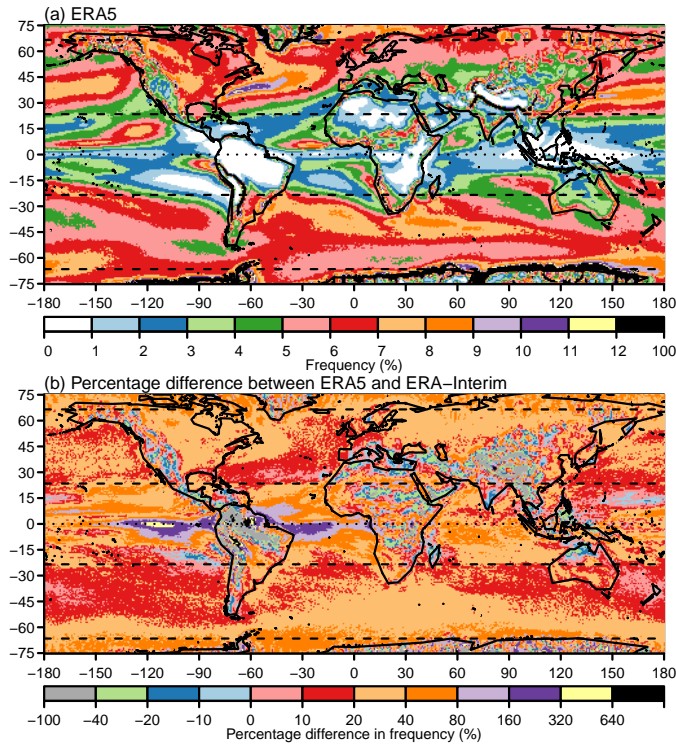

**Figure 8.** ERA5 compared to ERA-Interim. (a) ERA5 climatology of all fronts at $0.75° \times 0.75°$, and (b) percentage difference between ERA5 and ERA-Interim. The ERA5 climatology were computed with $n = 96$ smoothing cycles, $K_1 = -1.6 \times 10^{-11}\,\mathrm{K\,m^{-2}}$ and $K_2 = 7.5 \times 10^{-6}\,\mathrm{K\,m^{-1}}$ and $K_3 = 1.5\,\mathrm{m\,s^{-1}}$. ERA5 fronts were identified at $0.25° \times 0.25°$ then regridded to $0.75° \times 0.75°$ for comparison with ERA-Interim.

### 4.3  Front climatology from ERA5

The ERA5 reanalysis has a higher resolution than ERA-Interim, with grid spacing of $0.25° \times 0.25°$ compared to $0.75° \times 0.75°$ for ERA-Interim. For ERA5, a total of $n = 96$ smoothing cycles were required to make the climatologies of the TFP and gradient similar to ERA-Interim. Figure 3d illustrates fronts identified over Europe and the North Atlantic at 2001-01-01 00:00 UTC. As expected, the features are very similar to those identified in ERA-Interim against which it was calibrated (Figure 3c). Figure 8 compares the frequency of fronts identified in ERA5 with that in ERA-Interim when fronts are identified in ERA5 at $0.25° \times 0.25°$ grid spacing with $n = 96$ smoothing cycles but identical thresholds to those used for ERA-Interim, then aggregated to $0.75° \times 0.75°$ grid spacing for comparison with ERA-Interim. Aggregation is performed by counting individual fronts identified at the higher resolution passing through the lower resolution grid. When aggregated to the same resolution, more fronts are identified almost everywhere in ERA5 than in ERA-Interim. Since aggregation is performed by counting individual fronts, this indicates that ERA5 is able to resolve more fronts due to its higher resolution. The pattern of increase broadly follows the general distribution of fronts, with more fronts seen where they were already common, particularly in the storm tracks where front frequency increases by between $20\,\%$ and $40\,\%$. The greatest percentage increases are seen in the ITCZ

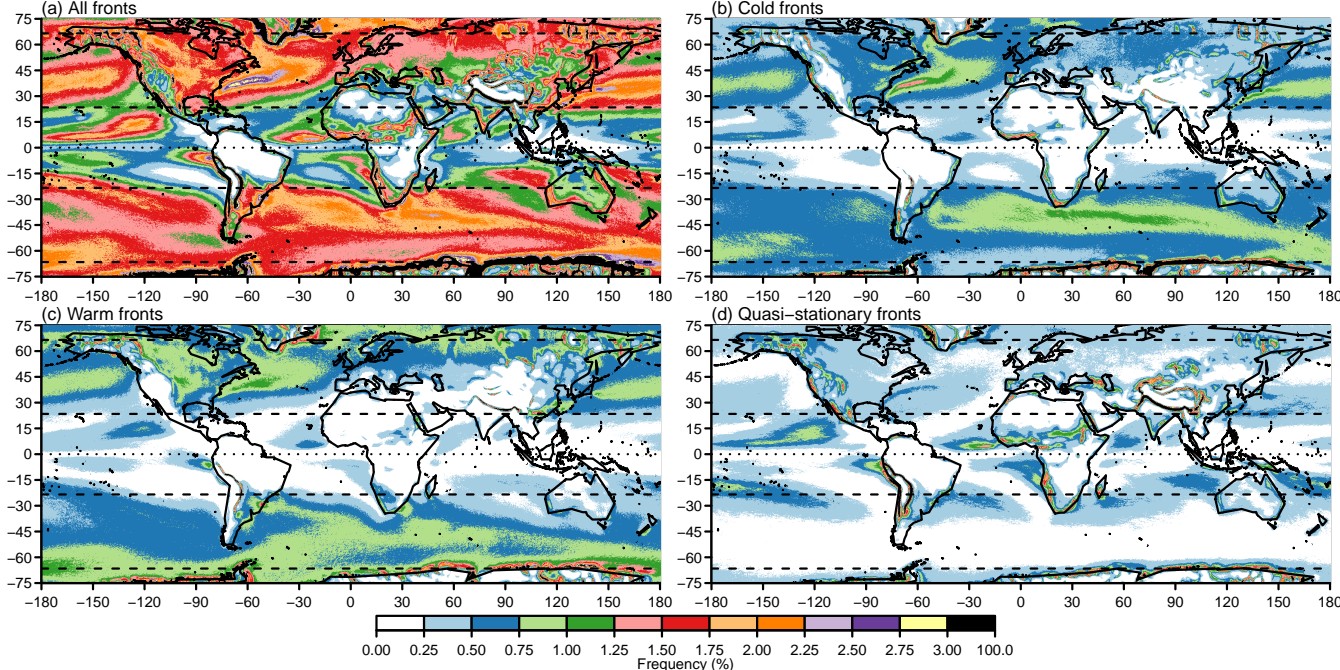

**Figure 9.** ERA5 global climatologies. (a) All fronts, (b) cold fronts, (c) warm fronts, (d) quasi-stationary fronts. All climatologies were computed with $n = 96$ smoothing cycles, $K_1 = -1.6 \times 10^{-11}\,\mathrm{K\,m^{-2}}$ and $K_2 = 7.5 \times 10^{-6}\,\mathrm{K\,m^{-1}}$ and $K_3 = 1.5\,\mathrm{m\,s^{-1}}$.

region to the east and west of South America where very few fronts were identified in ERA-Interim (and therefore represents a very small absolute increase in frequency), and is mostly associated with quasi-stationary fronts (Figure 9d.) Decreases in
325 frequency are primarily associated with areas of high orography, likely associated with the improved representation of the orography in the higher resolution dataset.

Figure 9 shows the climatology of fronts by type identified in ERA5 at its native $0.25° \times 0.25°$ resolution. Due to the smaller grid boxes, the frequency is necessarily lower than for ERA-Interim in Figure 6 and the aggregated data in Figure 8 One ERA-Interim grid box contains nine ERA5 grid boxes. A perfectly straight front passing through one ERA-Interim grid box would
330 typically pass through only three of the nine associated ERA5 grid boxes. Therefore, one might expect the front frequency in ERA5 at its native resolution to be approximately one third of the frequency in ERA-Interim. Comparing Figures 6 and 9 shows that this is approximately the case.

Figure 10 shows the seasonal breakdown of cold and warm fronts in ERA5, which is provided to be able to compare the most up-to-date climatology from ERA5 with previous studies. In general the maximum warm front frequency occurs at higher
335 latitudes than the maximum cold front frequency, due to the structure of extratropical cyclones and the associated poleward transport of warm air. During DJF especially, the high frequencies of atmospheric fronts that are influenced by the sea surface temperature (SST) fronts associated with the Gulf Stream in the North Atlantic and Kuroshio current in the North Pacific are

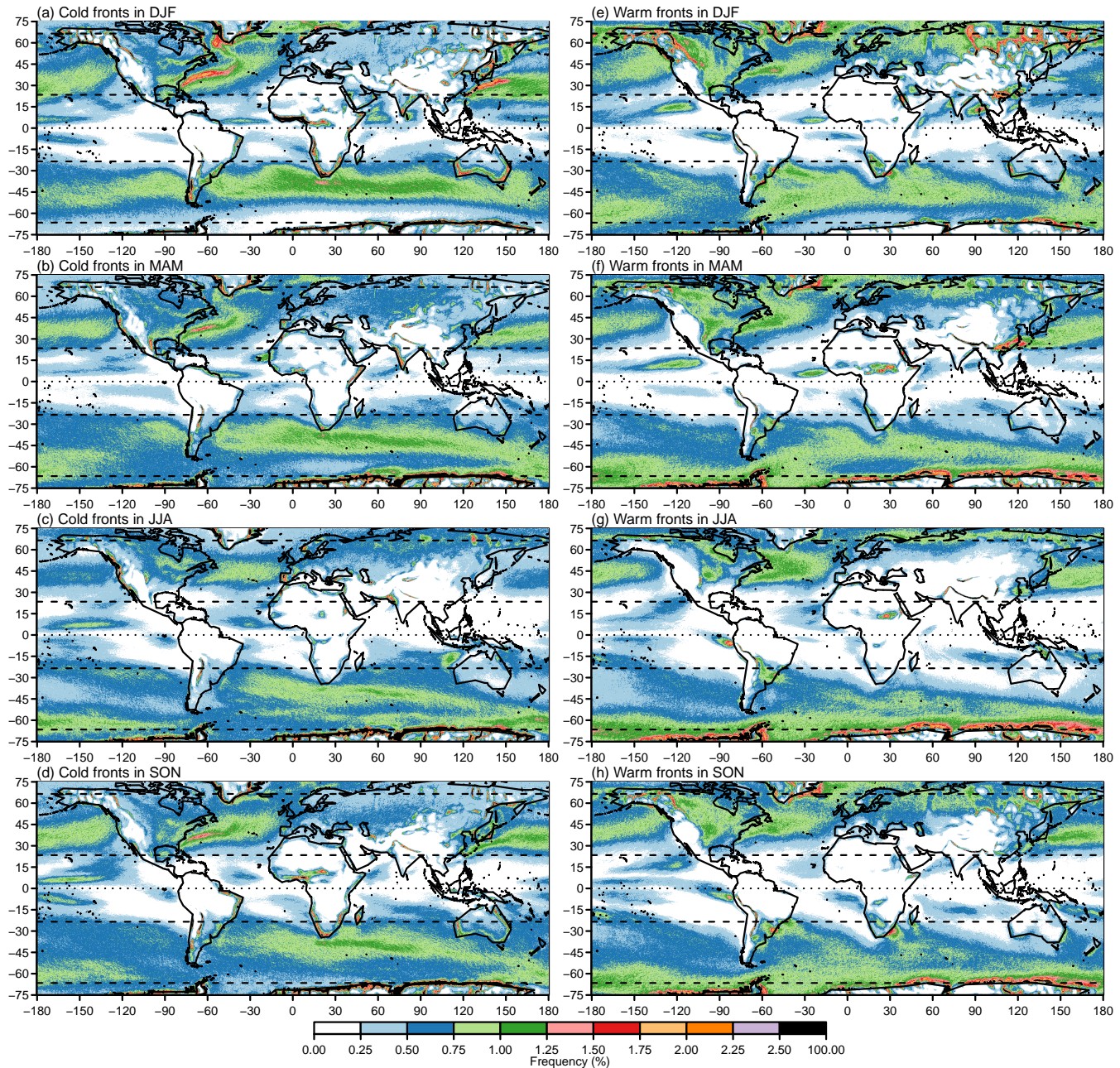

**Figure 10.** ERA5 seasonal climatologies of front frequency. Cold fronts (a-d), and warm fronts (e-h) for (a, e) DJF, (b, f) MAM, (c, g) JJA, (d, h) SON. All climatologies were computed with $n = 96$ smoothing cycles, $K_1 = -1.6 \times 10^{-11}\,\mathrm{K\,m^{-2}}$ and $K_2 = 7.5 \times 10^{-6}\,\mathrm{K\,m^{-1}}$ and $K_3 = 1.5\,\mathrm{m\,s^{-1}}$.

clearly visible. The influence of the SST on the atmosphere is more marked for higher resolution ocean and atmosphere (Parfitt et al., 2016, 2017a).

## 5 Discussion

In this paper, we have presented an updated implementation of the automatic front identification method of Berry et al. (2011b), based on H98. The updated implementation was designed specifically to scale to modern high resolution data sets. It is open source and does not require compilation, making it extremely portable. Despite not requiring compilation, computational performance of the new implementation in R is improved over earlier versions that were implemented in NCL with compiled components. Performance improvements come primarily from three areas: (i) the improved efficiency of the contouring algorithm compared to the line joining algorithm; (ii) vectorization of many calculations to avoid unnecessary loops; (iii) reduced memory usage by avoiding pre-allocating unnecessarily large arrays. One month of global ERA-Interim data at 6-hourly intervals and $0.75° \times 0.75°$ resolution can be processed in around 6 minutes using a single core of an Intel i7-8565U based laptop with a theoretical maximum speed of $4.6\,\mathrm{GHz}$. The same amount of global ERA5 data at $0.25° \times 0.25°$ can be processed in around 1 hour. Memory requirements are minimal since only one time step is processed at once. The improved scalability enables us to present high resolution climatologies of cold, warm and quasi-stationary fronts for all seasons from the ERA5 reanalysis.

In addition to several numerical improvements, the revised implementation uses the contour-then-mask approach originally proposed by H98 rather than the mask-then-join approach used by Berry et al. (2011b). The advantages of the contour-then-mask approach are demonstrated by example and by comparison of climatologies which show increased numbers of fronts identified almost everywhere. Gaps in what should be continuous fronts are reduced in ERA-Interim, and greater improvements are expected in lower resolution datasets for the reasons demonstrated in Figure 1. This improvement will be useful when linking frontal features to precipitation or winds (or compound extreme events) as in Catto and Dowdy (2021), or when using more object-based connections such as Papritz et al. (2014).

Most automatic feature detection algorithms require a calibration or training step involving comparison to analyses by a meteorologist. While this step cannot be neglected, we propose a quantile based approach to setting thresholds for front identification. Setting thresholds in terms of climatological quantiles makes the thresholds more easily comparable between data sets of differing resolution. By considering the climatological distributions of the masking variables, we have demonstrated for the first time the regional and seasonal variation of the TFP and gradient fields. Subsequent analyses may consider adopting latitudinally or seasonally varying thresholds in order to capture features that may be missed by or eliminate spurious features included by the used constant thresholds. In ERA5 this results in greater numbers of fronts identified even after smoothing, similar to the results of Parfitt et al. (2017b) after interpolation to lower resolution. Smoothing has the advantage of allowing feature identification to be conducted at the native resolution of each data set.

In addition to the various numerical and methodological improvements presented in this study, further numerical improvements, methodological choices, and alternative choices of meteorological field are possible. In addition to improving the accuracy of the finite difference approximations of the second derivative fields to second order, more accurate finite difference schemes could be used for both the first and second order derivatives. Following Berry et al. (2011b), we identify fronts as zero contours in the field defined by Equation 5 of H98, effectively the third derivative of the wet-bulb potential temperature

field at 850 hPa. Firstly, meteorological fields other than wet-bulb potential temperature could be considered, see H98 for a list of previously considered fields. Secondly, H98 derived an alternative expression for the front locator field, designed to reduce the frontal curvature. We retained the simpler definition for compatibility with Berry et al. (2011b) and subsequent studies, but the alternative definition preferred by H98 may be included as an option in a future version of the code associated with this study. Additional diagnostics such as distinguishing between between local and synoptic fronts suggested by Jenkner et al. (2010), or the additional criteria proposed by Hewson and Titley (2010) designed to eliminate spurious features associated with proximity to the warm conveyor belt, could also be implemented. Furthermore, while all distance calculations are carried out on a sphere in the updated implementation, contouring and interpolation still take place on a regular longitude-latitude grid. Greater accuracy could be achieved at high latitudes by also carrying out these operations on a sphere.

While cyclone identification algorithms routinely include the ability to track cyclonic features over subsequent time steps, similar feature tracking algorithms are almost absent for fronts. Front tracking is inherently more complex than cyclone tracking since fronts are complex line objects whereas cyclones can be reduced to simple point objects or point objects with an associated area. Hewson and Titley (2010) proposed a sophisticated tracking scheme for cyclonic features developing on fronts, which relies on accurate identification and classification of fronts in order to identify cyclones early in their life cycle, but is limited to tracking point objects associated with cyclones rather than fronts themselves. To the authors' knowledge, only Rüdisühli et al. (2020) have documented a front tracking algorithm. An openly available front tracking algorithm would offer new possibilities in terms of attributing and analysing impacts of individual fronts, e.g., precipitation or wind events, or understanding biases in weather and climate models.

*Code availability.* Code for revised method detailed in this paper are available from https://doi.org/10.5281/zenodo.7278068 and future developments will be available at https://github.com/phil-sansom/front_id.

*Author contributions.* PGS developed and tested the software, produced the results, and wrote the paper. JLC led the project, interpreted results, and wrote the paper.

*Competing interests.* The authors declare that they have no conflict of interest.

*Acknowledgements.* This research was supported by Natural Environment Research Council (NERC) grant NE/V004166/1. The authors thank Dr. Duncan Ackerley for comments on the manuscript.

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
