# Peer review of "Objective identification of meteorological fronts and climatologies"

_Geoscientific Model Development, 2022_

## Referee Comment (RC1)

**Review: Improved objective identification of meteorological fronts: a case study with ERA-Interim**

GMD submission 2022-255 by P. Sansom and J. Catto

The authors present an updated version of an established automated front identification method geared toward reanalysis datasets that are available with ever higher spatial resolution, along with an objective calibration method for the algorithm to make the identified fronts comparable between datasets with different spatial resolutions. They make a convincing case that their adaptations improve the efficacy of the algorithm, especially for moderate-resolution datasets like ERA-Interim that are still widely used, as well as climate models run at comparable resolutions. Given the high degree of subjectivity involved in automated front identification methods (choices of variables, of the degree of smoothing, of parameter thresholds), any successful effort to introduce more objectivity is welcome, and the presented calibration method appears to work well for the datasets in question. The manuscript is well structured and well written, and in my opinion only requires minor adaptations. I therefore recommend it for publication pending minor revisions.

Following is a list of questions, requests and recommendations, roughly grouped into major, Figure-related and minor points.

**Major**

1. Lines 1—11: Mention reanalysis datasets in the Abstract and quantify the improvements or changes over the old method (e.g., change in no. fronts overall and in storm track regions).
2. Lines 40—50: "This can impact the attribution of precipitation to fronts." How?
3. Line 112: What does "joins between adjacent grid boxes" mean? Consider reformulating.
4. Line 117: I understand "even moderately high resolution" as "moderately high or higher resolution" (i.e., as "low-to-high-resolution"), when what appears to be meant is the opposite, "moderately high or lower resolution" (i.e., "high-to-low-resolution"). Please reformulate to remove this unambiguity.
5. Line 125: "More fronts" despite the fact that closing gaps between partial fronts decreases the overall number of front objects? Is this increase overcompensated by the fact that such joined fronts meet the minimum length criterion, which the partial fronts identified by the old approach might not? Or are there other factors at play?
6. Line 149: Why do you choose the $25^{th}$ percentile specifically, and not any other percentile below the $50^{th}$? Does it fit the data especially well? And how objective is this choice?
7. Line 150: Why is the $50^{th}$ percentile 0 K m-2; does this just happen to be the case by chance, or is there some theoretical reason for this symmetry around zero?
8. Line 153: What do you mean by "simple symmetry"?
9. Line 164: Are the climatologies required to determine the quantiles? Please state this.
10. Lines 169—170: Which month did you end up using? Does it make a difference which month/season? How did you reach this conclusion, i.e., what did you test?

11. Line 174: Do I correctly understand that "object" refers to the front and for a zonally oriented front, you use the grid box to the left and that to the right of it? What happens for diagonally oriented fronts; do you use the top-left and the bottom-right neighboring boxes in that case? This could be formulated a bit more clearly.

12. Lines 173/175—176: What is the difference between the "simple central finite difference" and "explicit central finite difference" approximations? A short explanation could be helpful (unless these terms are obvious to people more versed in numerical discretization than I am).

13. Line 180: Add reference for NCL.

14. Line 196: Briefly mention Figure 4d to complete the discussion of the figure, e.g., "Figure 4d shows the fronts in ERA5 and will be discussed in Section X."

15. Lines 256—257: Why is the performance improved over earlier versions? Does the new version maybe use optimized third-party libraries over handwritten code? Please add a short explanation.

16. Line 258: I very much appreciate the performance numbers, but please replace "a single core of a 2 year old laptop" by a more meaningful hardware description (processor type, number of cores etc.).

17. Line 268: Why is contouring cheaper than line joining? If possible, add a very brief explanation.

18. Line 279: Consider replacing "performance" by a synonym when describing the efficacy or accuracy of the identification algorithm, as "performance" often refers to the compute performance (which, in this case, would likely rather be decreased than increased by moving to higher-order accuracy).

19. Line 286: What kind of performance benefit do you refer to, compute performance or the performance (accuracy) of the algorithm?

**Figures**

20. Figure 1, panels: Consider adding labels above the plots, e.g., "(a) Mask-then-join" and "(b) Contour-then-mask".

21. Figure 1, caption: Merge first two sentences; consider removing "Thick" before "black" (there are no thin contours other than country borders).

22. Figure 1, caption: Consider changing "black lines are contours of the" to the, e.g., "black contours show" (shorter; likewise in other captions).

23. Figure 1, caption: Replace "Points" by "Circles".

24. Figure 2, plots: Add a legend for the line colors.

25. Figure 2, panels: Consider adding labels above the plots, e.g., "(a) Zonal-mean TFP" and "(b) Zonal-mean |nabla TH_W|".

26. Figure 2, plots: The yellowish lines are close to invisible on my (full-color) printout, and most likely hard to see on many projectors; consider adapting the colors.

27. Figure 2, caption: Add the threshold values.

28. Figure 3, panels: Add labels above the panels, e.g., "(a) Mask-then-join", "(b) Contour-then-mask (n=2)", "(c)Contour-then-mask (n=8)", "(d) ERA5".

29. Figure 3, caption: Mention that (a—c) are based on ERA-Interim data.

30. Figure 3, caption: Add the date and time of the snapshot.

31. Figures 4—10, plots: Use a different colormap for frequency plots; on the on hand, one comprised of multiple colors (not just shades of red) would make regional differences stand

out more (especially in Figure 7 and 10, where all panels look the same at first glance) and increase the visual appeal of the maps; on the other hand, using the same shades of red for absolute frequencies and positive differences makes it hard to distinguish these two types of plots.

32. Figure 4, panels: Add labels, e.g., "(a) Dowdy and Catto (2017)", "(b) Mask-then-join", "(c) Contour-then-mask", "(c) Difference (b) – (a)", "(d) Difference (c) – (a)", "(f) Difference (c) – (b)".
33. Figure 5, plot: Fix overlapping colorbar labels (here and throughout the manuscript).
34. Figure 6, panels: Add labels, e.g., "(a) Cold fronts" etc.
35. Figure 6, panels: Reorder the panels such that the total front frequency comes first (top-left), which is more intuitive, especially given (d) is discussed first in the text before (a—c); the panels of a figure should ideally be discussed in order in the text, which reordering would achieve.
36. Figure 7, panels: Add labels above plots, e.g., "(a) Cold fronts in DJF" etc.
37. Figure 8, panels: Add labels above plots, e.g., "(a) ERA5", "(b) Difference ERA5 - ERA-Interim".
38. Figure 9, panels: Add labels and reorder panels as in Figure 6.
39. Figure 10, panels: Add labels above plots, e.g., "(a) Cold fronts in DJF" etc.

**Minor**

40. Line 2: Consider adding "weather prediction" after "operational".
41. Lines 4—5: Sentence is a bit hard to read as the two approaches ("applying a mask then joining frontal points" and "contouring the thermal field then applying the mask") don't stand out very well; could be improved by, e.g., adding "(i)" and "(ii)" or so before "applying" and "contouring", respectively.
42. Line 6: Add "for" after "allows".
43. Line 7: Consider replacing "have made" by "present" or similar.
44. Line 23: Remove "on" after "impact".
45. Line 25: Remove "prior to then" (he would have been hard pressed to summarized future methods).
46. Lines 26—29: Revise "Tomas […] thresholds": Several incomplete sentences that could be merged into one (e.g., "[…] parameter: first, […] 850 hPa; second, […] derivative; and […]").
47. Line 31: Remove "a thermal variable" (the whole paragraph is about approaches based on thermal variables) and remove parentheses around "equivalent potential temperature".
48. Line 31: Replace "the second derivative of that variable" by "its second derivative".
49. Line 33: Replace "which" by "who".
50. Line 34: Remove "and" before "placing" or replace the latter by "placed".
51. Line 40: Is "benefiting" really the right word here?
52. Line 54: Move "Hewson" out of parentheses (probably used "\citep" instead of "\citet" in LaTeX source code).
53. Line 55: Consider streamlining "We demonstrate a method that can be used […]".
54. Lines 61—62: Add "the" before "European and consider moving "reanalysis" after "ERA-Interim"; define "ERA-Interim"; consider putting the definitions of ECMWF and ERA-Interim in parentheses instead of the acronyms (easier to read than vice versa).
55. Line 64: Remove "The wet bulb potential temperature TH_W is computed" and join the sentence with the previous one ("[…] identify fronts, using the […]").

56. Line 67: Define ERA5 and move reference here from line 71.
57. Line 68: Add "of," "among" or similar before "standard" (something is missing there).
58. Line 71: Reference ERA5 at first occurrence on line 67.
59. Line 77: Are both equations necessary, i.e., is it not obvious that "nabla times nabla" is the same as "nabla squared"?
60. Line 79: As far as I can tell, "Thermal Front Locator" should not be written in uppercase.
61. Line 83: Add comma before "where".
62. Lines 91—93: Something appears to be missing from this sentence (around "K_2 a fraction of a grid length").
63. Line 100: Consider replacing "1, 2, 3 and 4" by "1—4".
64. Line 101: Consider adding after "improvement" something along the lines of "described in the next section".
65. Line 110: What does "simultaneously" refer to? If it means that zero points are identified in parallel, then reformulate accordingly. (Grammatically, it could also refer to identifying points and joining them into lines in parallel, but that doesn't make sense as far as I can tell.)
66. Line 135: Replace "smooth [features]" by "continuous" (provided I understand the sentence correctly).
67. Line 136: Consider joining the sentences, e.g., "[…] Equation 1, which will appear […]".
68. Lines 137—138: Consider replacing "the contours of TFL = 0 K m-3" by "these contours".
69. Line 139: Consider adding "filter" after "average" and "data" or "fields" after "ERA-Interim".
70. Line 146: What does "gradient" refer to?
71. Line 146: Consider adding "the" before "gradient".
72. Line 147: Replace "very [different]" by, e.g., "more variable and different" ("variable" in contrast to "constant" in the extratropics, and "different" because the values differ).
73. Line 157: If "previously" refers to Berry et al., replace it by "their study", "Berry et al." or the full reference.
74. Line 160: Consider replacing "the datasets of interest" by "them".
75. Lines 163—164: Replace "at [any reasonable]" by "for" ("at" doesn't fit to "number of smoothing passes") and "or [number]" by "and".
76. Line 165: Replace "so [that]" by "such".
77. Line 166: I think "magnitude" should be plural.
78. Line 168: By "can be compared […] only for", do you mean that it is not possible to compare the percentile for a larger subset of the data (let alone the whole datasets), or do you mean that it is sufficient to only use a small subset of each dataset to meaningfully compare them? "Can be […] only" reads like the former (i.e., a limitation), but judging from "avoiding the need […]", I think you mean the latter (i.e., an opportunity). Please reformulate to clarify this.
79. Line 168: Add "that of" before "the gradient field".
80. Line 172: Consider swapping "single" and "biggest".
81. Line 174: Add "on" before "either side".
82. Line 184: Replace "criteria" by "criterion".
83. Line 190: Consider replacing "smoothing passes" by "n" for consistency with "K_1" and "K_2" (and add a comma after "i.e."); add "UTC" after "00:00" (or after the date).
84. Lines 198—199: Join sentences with comma after "i.e., [add comma] the […] K_2" or make the second sentence ("Specifically, […]") complete by adding a verb.
85. Lines 215—216: Consider shortening the title by removing "identified" and maybe even "data"; alternatively, replace "Fronts identified" by "front climatology" to distinguish the

section from those before, which already discussed fronts in ERA-Interim but focused on different identification approaches rather than on the climatologies.

86. Line 217: Add "for" after "allows".
87. Line 225: Remove "in [in]".
88. Lines 225—229: Put all figure references in parentheses instead of writing "in Figure X".
89. Line 230: Same as line 215.
90. Line 233: Add "UTC" after "00:00" (or after the date).
91. Lines 234—235: Replace "[calibrated] in Figure 3(c)" by "(Figure 3c)".
92. Line 235: Add "that in" or similar before "ERA-Interim" (the frequency of fronts in ERA5 is not compared with ERA-Interim itself, but with the frequency of fronts in ERA-Interim).
93. Line 236: What are the thresholds identical to?
94. Line 240: Replace "[fronts] seen where there fronts [were]" by "where they".
95. Line 242: Replace "[at] it's [native]" by "its".
96. Line 243: Add "Figure" before "[in] 8".
97. Line 249: Consider replacing "associated" with a more suitable word (it feels a bit off to me).
98. Lines 262—263: Reformulate to avoid using "contour-then-mask approach" twice in close succession.
99. Line 269: Reformulate to avoid using "involve" and "involving" in close succession.
100. Line 270: I understand the usage of "human" to highlight the subjectivity introduced by a meteorologist's expert judgement, but since I am not aware of the existence of nonhuman meteorologists (weather-sensitive animals aside), please consider reformulating this or simply removing "human".
101. Line 277: "Alternative choices" of what?
102. Line 277: Replace "As well as" by, e.g., "In addition to".
103. Line 280: "Implied" is probably a typo and should be "increased" or similar.
104. Line 285: Consider replacing "6, 7 & 8" by "6—8" (or at least "6, 7, 8") and "Equations 6—8 of that paper" by "their Equations 6—8".
105. Line 288: Add some separator before "while", e.g., "Furthermore, [while]", to gain some separation from the previous, unrelated sentence.

---

## Author Comment (AC1)

**Response to Referee 1 for GMD-2022-255**

Philip G. Sansom and Jennifer L. Catto

December 2023

**Referee 1**

The authors present an updated version of an established automated front identification method geared toward reanalysis datasets that are available with ever higher spatial resolution, along with an objective calibration method for the algorithm to make the identified fronts comparable between datasets with different spatial resolutions. They make a convincing case that their adaptations improve the efficacy of the algorithm, especially for moderate-resolution datasets like ERA-Interim that are still widely used, as well as climate models run at comparable resolutions. Given the high degree of subjectivity involved in automated front identification methods (choices of variables, of the degree of smoothing, of parameter thresholds), any successful effort to introduce more objectivity is welcome, and the presented calibration method appears to work well for the datasets in question. The manuscript is well structured and well written, and in my opinion only requires minor adaptations. I therefore recommend it for publication pending minor revisions.

Following is a list of questions, requests and recommendations, roughly grouped into major, Figure-related and minor points.

*The authors thank the reviewer for their detailed comments which are addressed individually below.*

**Major**

1. Lines 1—11: Mention reanalysis datasets in the Abstract and quantify the improvements or changes over the old method (e.g., change in no. fronts overall and in storm track regions).

*We have added mention of reanalysis datasets. The improvements are detailed in terms of the smoothness of the fronts and fewer breaks for example.*

2. Lines 40—50: "This can impact the attribution of precipitation to fronts." How?

*Clarified in text: "This was shown lead to large differences between datasets in the proportion of precipitation attributed to fronts."*

3. Line 112: What does "joins between adjacent grid boxes" mean? Consider reformulating.

*Clarified in text: "but only zero points located in adjacent grid boxes are considered for joining into lines"*

4. Line 117: I understand "even moderately high resolution" as "moderately high or higher resolution" (i.e., as "low-to-high-resolution"), when what appears to be meant is the opposite, "moderately high or lower resolution" (i.e., "high-to-low-resolution"). Please reformulate to remove this unambiguity.

*Clarified in text: "At or below the $0.75° \times 0.75°$ resolution of ERA-Interim"*

5. Line 125: "More fronts" despite the fact that closing gaps between partial fronts decreases the overall number of front objects? Is this increase over-compensated by the fact that such joined fronts meet the minimum length criterion, which the partial fronts identified by the old approach might not? Or are there other factors at play?

*A combination of increased lengths meeting the minimum length criteria, and the identification of front points and fronts missed completely by the old method, see for Figure 3 for examples. Clarified in text.*

6. Line 149: Why do you choose the 25th percentile specifically, and not any

other percentile below the 50th ? Does it fit the data especially well? And how objective is this choice?

*Comparison with charts and ability to produce continuous fronts. A range of combinations of thresholds $K_1$ and $K_2$ were explored and now shown in the supplementary material, but ultimately the choice is still subjective. Clarified in text.*

7. Line 150: Why is the 50th percentile 0 K m-2; does this just happen to be the case by chance, or is there some theoretical reason for this symmetry around zero?

*The TFP is essentially the second derivative of $\theta_W$, so intuitively it makes sense that over a large area the average should be approximately zero, but over smaller areas or time scales, it need not be.*

8. Line 153: What do you mean by "simple symmetry"?

*Each is approximately the 50th percentile of the allowed range, since we require $K_1 < 0$. Clarified in text.*

9. Line 164: Are the climatologies required to determine the quantiles? Please state this.

*Clarified at the beginning of paragraph 3 of Section 3.2.*

10. Lines 169—170: Which month did you end up using? Does it make a difference which month/season? How did you reach this conclusion, i.e., what did you test?

*Various lengths of training data and spatial extents were tested from one month to 30 years. In practice we used January 2000 for the Norther Hemisphere extra-tropics. Clarified in text.*

11. Line 174: Do I correctly understand that "object" refers to the front and for a zonally oriented front, you use the grid box to the left and that to the right of it? What happens for diagonally oriented fronts; do you use the top-left and the bottom-right neighboring boxes in that case? This could be formulated a bit more clearly.

*Object refers to a grid point, Equations 1–4 are evaluated at each grid point, then interpolated to front points. The zonal and meridional components of the derivatives are computed separately using one grid box to the left and right, or above and below respectively. Clarified in text.*

12. Lines 173/175—176: What is the difference between the "simple central finite difference" and "explicit central finite difference" approximations? A short explanation could be helpful (unless these terms are obvious to people more versed in numerical discretization than I am).

*In the original code, approximations to the second and third derivatives required in Equations 1-4 were obtained by repeated applications of the "simple" finite difference approximation to the first derivative. We use an explicit finite difference approximation to the second derivative in order to evaluate $\nabla^2$, avoiding the need for repeated applications of the "simple" approximation which degrade accuracy. Clarified in text.*

13. Line 180: Add reference for NCL.

*Done.*

14. Line 196: Briefly mention Figure 4d to complete the discussion of the figure, e.g., "Figure 4d shows the fronts in ERA5 and will be discussed in Section X."

*Figure 3d, Done.*

15. Lines 256—257: Why is the performance improved over earlier versions? Does the new version maybe use optimized third-party libraries over hand-written code? Please add a short explanation.

*Clarified in text: "Performance improvements come primarily from three areas: (i) the improved efficiency of the contouring algorithm compared to the line joining algorithm; (ii) vectorization of many calculations to avoid unnecessary loops; (iii) reduced memory usage by avoiding pre-allocating unnecessarily large arrays."*

16. Line 258: I very much appreciate the performance numbers, but please replace "a single core of a 2 year old laptop" by a more meaningful hardware

description (processor type, number of cores etc.).

*Clarified in text: "a single core of an Intel i7-8565U based laptop with a theoretical maximum speed of 4.6 GHz"*

17. Line 268: Why is contouring cheaper than line joining? If possible, add a very brief explanation.

*This was explained in Section 3.1, where it has been expanded.*

18. Line 279: Consider replacing "performance" by a synonym when describing the efficacy or accuracy of the identification algorithm, as "performance" often refers to the compute performance (which, in this case, would likely rather be decreased than increased by moving to higher-order accuracy).

*Clarified in text: "modest increases in both the number of fronts and front points identified"*

19. Line 286: What kind of performance benefit do you refer to, compute performance or the performance (accuracy) of the algorithm?

*This comment has been removed in response to comments from Reviewer 2. A more complete comparison would be enlightening, but is beyond the scope of the current study.*

**Figures**

20. Figure 1, panels: Consider adding labels above the plots, e.g., "(a) Mask-then-join" and "(b) Contour-then-mask".

*Done.*

21. Figure 1, caption: Merge first two sentences; consider removing "Thick" before "black" (there are no thin contours other than country borders).

*Done.*

22. Figure 1, caption: Consider changing "black lines are contours of the"

to the, e.g., "black contours show" (shorter; likewise in other captions).

*Done.*

23. Figure 1, caption: Replace "Points" by "Circles".

*Done.*

24. Figure 2, plots: Add a legend for the line colors.

*Done.*

25. Figure 2, panels: Consider adding labels above the plots, e.g., "(a) Zonal-mean TFP" and "(b) Zonal-mean $|\nabla\theta_W|$".

*Done.*

26. Figure 2, plots: The yellowish lines are close to invisible on my (full-color) printout, and most likely hard to see on many projectors; consider adapting the colors.

*Done.*

27. Figure 2, caption: Add the threshold values.

*Done.*

28. Figure 3, panels: Add labels above the panels, e.g., "(a) Mask-then-join", "(b) Contour-then-mask (n=2)", "(c)Contour-then-mask (n=8)", "(d) ERA5".

*Done.*

29. Figure 3, caption: Mention that (a—c) are based on ERA-Interim data.

*Done.*

30. Figure 3, caption: Add the date and time of the snapshot.

*Done.*

31. Figures 4—10, plots: Use a different colormap for frequency plots; on the on hand, one comprised of multiple colors (not just shades of red) would make regional differences stand out more (especially in Figure 7 and 10, where all panels look the same at first glance) and increase the visual appeal of the maps; on the other hand, using the same shades of red for absolute frequencies and positive differences makes it hard to distinguish these two types of plots.

*Done.*

32. Figure 4, panels: Add labels, e.g., "(a) Dowdy and Catto (2017)", "(b) Mask-then-join", "(c) Contour-then-mask", "(c) Difference (b) – (a)", "(d) Difference (c) – (a)", "(f) Difference (c) – (b)".

*Done.*

33. Figure 5, plot: Fix overlapping colorbar labels (here and throughout the manuscript).

*Done.*

34. Figure 6, panels: Add labels, e.g., "(a) Cold fronts" etc.

*Done.*

35. Figure 6, panels: Reorder the panels such that the total front frequency comes first (top-left), which is more intuitive, especially given (d) is discussed first in the text before (a—c); the panels of a figure should ideally be discussed in order in the text, which reordering would achieve.

*Done.*

36. Figure 7, panels: Add labels above plots, e.g., "(a) Cold fronts in DJF" etc.

*Done.*

37. Figure 8, panels: Add labels above plots, e.g., "(a) ERA5", "(b) Difference ERA5 - ERA- Interim".

*Done.*

38. Figure 9, panels: Add labels and reorder panels as in Figure 6.

*Done.*

39. Figure 10, panels: Add labels above plots, e.g., "(a) Cold fronts in DJF" etc.

*Done.*

**Minor**

40. Line 2: Consider adding "weather prediction" after "operational".

*Done.*

41. Lines 4—5: Sentence is a bit hard to read as the two approaches ("applying a mask then joining frontal points" and "contouring the thermal field then applying the mask") don't stand out very well; could be improved by, e.g., adding "(i)" and "(ii)" or so before "applying" and "contouring", respectively.

*This sentence has been reformulated in response to comments from Reviewer 2.*

42. Line 6: Add "for" after "allows".

*Done.*

43. Line 7: Consider replacing "have made" by "present" or similar.

*Done.*

44. Line 23: Remove "on" after "impact".

*Done.*

45. Line 25: Remove "prior to then" (he would have been hard pressed to summarized future methods).

*Done.*

46. Lines 26—29: Revise "Tomas [. . . ] thresholds": Several incomplete sentences that could be merged into one (e.g., "[. . . ] parameter: first, [. . . ] 850 hPa; second, [. . . ] derivative; and [. . . ]").

*Done.*

47. Line 31: Remove "a thermal variable" (the whole paragraph is about approaches based on thermal variables) and remove parentheses around "equivalent potential temperature".

*Done.*

48. Line 31: Replace "the second derivative of that variable" by "its second derivative".

*Done.*

49. Line 33: Replace "which" by "who".

*Done.*

50. Line 34: Remove "and" before "placing" or replace the latter by "placed".

*Done.*

51. Line 40: Is "benefiting" really the right word here?

*Replaced by "tailored to suit".*

52. Line 54: Move "Hewson" out of parentheses (probably used "citep" instead of "citet" in LaTeX source code).

*Done.*

53. Line 55: Consider streamlining "We demonstrate a method that can be used [. . . ]".

*Done.*

54. Lines 61—62: Add "the" before "European and consider moving "reanalysis" after "ERA- Interim"; define "ERA-Interim"; consider putting the definitions of ECMWF and ERA-Interim in parentheses instead of the acronyms (easier to read than vice versa).

*Done, but elsewhere since first references to these have been made earlier.*

55. Line 64: Remove "The wet bulb potential temperature $\theta_W$ is computed" and join the sentence with the previous one ("[...] identify fronts, using the [...]").

*Done.*

56. Line 67: Define ERA5 and move reference here from line 71.

*Done.*

57. Line 68: Add "of," "among" or similar before "standard" (something is missing there).

*Added "among".*

58. Line 71: Reference ERA5 at first occurrence on line 67.

*Done.*

59. Line 77: Are both equations necessary, i.e., is it not obvious that $\nabla \times \nabla$ is the same as $\nabla^2$?

$\nabla^2$ *is sometimes also used to indicate the Hessian matrix, so we have retained both equations for clarity, and compatibility with Hewson [1998].*

60. Line 79: As far as I can tell, "Thermal Front Locator" should not be written in uppercase.

*Changed to lower case.*

61. Line 83: Add comma before "where".

*Done.*

62. Lines 91—93: Something appears to be missing from this sentence (around "$K_2$ a fraction of a grid length").

*Clarified - "at a point a fraction of grid length"*

63. Line 100: Consider replacing "1, 2, 3 and 4" by "1—4".

*Done.*

64. Line 101: Consider adding after "improvement" something along the lines of "described in the next section".

*Done - "described in Section 3.1"*

65. Line 110: What does "simultaneously" refer to? If it means that zero points are identified in parallel, then reformulate accordingly. (Grammatically, it could also refer to identifying points and joining them into lines in parallel, but that doesn't make sense as far as I can tell.)

*Removed. In the algorithm used by Berry et al. [2011] locating zero points and joining them into lines were two separate steps, which in our implementation are done by a single function call. The more important performance distinction has been clarified in the next sentence.*

66. Line 135: Replace "smooth [features]" by "continuous" (provided I understand the sentence correctly).

*Done.*

67. Line 136: Consider joining the sentences, e.g., "[...] Equation 1, which will appear [...]".

*Done.*

68. Lines 137—138: Consider replacing "the contours of TFL = 0 K m-3" by "these contours".

*Done.*

69. Line 139: Consider adding "filter" after "average" and "data" or "fields"

after "ERA-Interim".

*Added "data" after ERA-Interim.*

70. Line 146: What does "gradient" refer to?

*Equation 3, clarified in text.*

71. Line 146: Consider adding "the" before "gradient".

*Done.*

72. Line 147: Replace "very [different]" by, e.g., "more variable and different" ("variable" in contrast to "constant" in the extratropics, and "different" because the values differ).

*This paragraph has been reformulated in response to comments from Reviewer 2.*

73. Line 157: If "previously" refers to Berry et al., replace it by "their study", "Berry et al." or the full reference.

*Added full reference.*

74. Line 160: Consider replacing "the datasets of interest" by "them".

*Done.*

75. Lines 163—164: Replace "at [any reasonable]" by "for" ("at" doesn't fit to "number of smoothing passes") and "or [number]" by "and".

*Done.*

76. Line 165: Replace "so [that]" by "such".

*Done.*

77. Line 166: I think "magnitude" should be plural.

*Don't believe so, rephrased.*

78. Line 168: By "can be compared [...] only for", do you mean that it is not possible to compare the percentile for a larger subset of the data (let alone the whole datasets), or do you mean that it is sufficient to only use a small subset of each dataset to meaningfully compare them? "Can be [...] only" reads like the former (i.e., a limitation), but judging from "avoiding the need [...]", I think you mean the latter (i.e., an opportunity). Please reformulate to clarify this.

*The latter, rephrased for clarity.*

79. Line 168: Add "that of" before "the gradient field".

*Done.*

80. Line 172: Consider swapping "single" and "biggest".

*This sentence has been removed.*

81. Line 174: Add "on" before "either side".

*Done.*

82. Line 184: Replace "criteria" by "criterion".

*Done.*

83. Line 190: Consider replacing "smoothing passes" by "n" for consistency with "$K_1$" and "$K_2$" (and add a comma after "i.e."); add "UTC" after "00:00" (or after the date).

*Done.*

84. Lines 198—199: Join sentences with comma after "i.e., [add comma] the [...] $K_2$)" or make the second sentence ("Specifically, [...]") complete by adding a verb.

*Joined sentences.*

85. Lines 215—216: Consider shortening the title by removing "identified" and maybe even "data"; alternatively, replace "Fronts identified" by

"front climatology" to distinguish the section from those before, which already discussed fronts in ERA-Interim but focused on different identification approaches rather than on the climatologies.

*Shortened to "Front climatology from ERA-Interim"*

86. Line 217: Add "for" after "allows".

*Done.*

87. Line 225: Remove "in [in]".

*Done.*

88. Lines 225—229: Put all figure references in parentheses instead of writing "in Figure X".

*Done.*

89. Line 230: Same as line 215.

*Shortened to "Front climatology from ERA5"*

90. Line 233: Add "UTC" after "00:00" (or after the date).

*Done.*

91. Lines 234—235: Replace "[calibrated] in Figure 3(c)" by "(Figure 3c)".

*Done.*

92. Line 235: Add "that in" or similar before "ERA-Interim" (the frequency of fronts in ERA5 is not compared with ERA-Interim itself, but with the frequency of fronts in ERA-Interim).

*Done.*

93. Line 236: What are the thresholds identical to?

*Clarified - "to those used for ERA-Interim"*

94. Line 240: Replace "[fronts] seen where there fronts [were]" by "where they".

*Done.*

95. Line 242: Replace "[at] it's [native]" by "its".

*Done.*

96. Line 243: Add "Figure" before "[in] 8".

*Done.*

97. Line 249: Consider replacing "associated" with a more suitable word (it feels a bit off to me).

*Replaced "associated with" with "due to"*

98. Lines 262—263: Reformulate to avoid using "contour-then-mask approach" twice in close succession.

*Reformulated.*

99. Line 269: Reformulate to avoid using "involve" and "involving" in close succession.

*Replaced "involve" with "require"*

100. Line 270: I understand the usage of "human" to highlight the subjectivity introduced by a meteorologist's expert judgement, but since I am not aware of the existence of nonhuman meteorologists (weather-sensitive animals aside), please consider reformulating this or simply removing "human".

*Removed "human".*

101. Line 277: "Alternative choices" of what?

*Meteorological field, clarified in text.*

102. Line 277: Replace "As well as" by, e.g., "In addition to".

*Done.*

103. Line 280: "Implied" is probably a typo and should be "increased" or similar.

*Replaced "implied" with "increased".*

104. Line 285: Consider replacing "6, 7 & 8" by "6—8" (or at least "6, 7, 8") and "Equations 6—8 of that paper" by "their Equations 6—8".

*Removed as part of changes in response to comments by Reviewer 2.*

105. Line 288: Add some separator before "while", e.g., "Furthermore, [while]", to gain some separation from the previous, unrelated sentence.

*Done.*

**References**

Gareth Berry, Michael J. Reeder, and Christian Jakob. A global climatology of atmospheric fronts. *Geophysical Research Letters*, 38(4):1–5, 2011. doi: 10.1029/2010GL046451.

T. D. Hewson. Objective fronts. *Meteorological Applications*, 5(1):37–65, 1998. ISSN 13504827. doi: 10.1017/S1350482798000553.

---

## Author Comment (AC2)

**Response to Referee 2 GMD-2022-255**

Philip G. Sansom and Jennifer L. Catto

December 2023

**Reviewer 2**

**Summary**

This paper looks into the issue of simplifying and speeding up objective front identification in re-analyses, with simple application across a range of numerical model types/resolutions in mind, to facilitate intercomparison of (e.g.) resolution impact. The goals are OK, but in my view they have not really been addressed or reached in any useful or useable way. And I do not agree that the main method chosen is the right one. In tandem, the paper has many other weak areas: textual clarity, figure clarity, disingenuous scale selection, result misinterpretation, unsubstantiated claims, etc. This all means that the manuscript does not unfortunately contain sufficient new science, new methods or new results to warrant publication in my view. With regard to the claimed innovations (e.g. in the abstract) these are either minor adjustments, or have been taken from another publication without acknowledgement. This also raises questions of scientific integrity, which is clearly disappointing. There are a few areas where, with further investigative work, a wholly restructured manuscript could potentially reach a publishable standard.

*The authors appreciate the detailed comments and insights provided by the reviewer and have addressed them in detail below. In particular, all figures*

*have been revised with new colour scales for increased clarity, and the scale selection clarified. The mistake in not correctly identifying that the alternative order of operations had already been proposed elsewhere has been corrected. However, we believe that identifying and demonstrating the difference compared to the previous implementation which had been used in multiple studies is still a valuable contribution. The goal of this study was to document a new portable, scalable, open-source software implementation of a widely used algorithm in a transparent manner, and to provide updated climatologies, including new high resolution climatologies. In addition we believe that the proposed quantile based definitions of the front identification thresholds will facilitate easier comparison between datasets of different resolutions, whether or not additional smoothing is applied to make those datasets more similar.*

**Detailed comments**

**Major Points**

1. In highlighting changing the 'order of operations' in the abstract you are merely copying the method of Hewson (1998) (his Fig 2), without acknowledging that. Similarly the recommendation to use a contouring algorithm (L111) is what Hewson (1998) proposed and implemented, and you don't acknowledge that either. So whilst these updates do deliver better results, the authors are wrong to imply it is their innovation. Then referencing 'more accurate finite differencing' in the abstract would mean, one would think, that this is something new and very different, whereas it is a second order centred finite difference, which to my mind would ordinarily be the default way to compute del-squared on a grid. This is also rather trivial compared to the extensive work done to compute derivatives correctly in Hewson (1998) – ref: P46 and appendices 1 and 2. So I can't really see how any of these aspects can be justifiably referenced in the abstract. Likewise the other methodological changes mentioned in the abstract - direct calculation of wet-bulb potential temperature and better handling of short fronts – are to my mind small adaptations that should appear only in the main text as minor points, and not in the abstract as if they represented major progress. So virtually all the "key points" of the abstract are

not key points at all, but either copies of previously published work or small algorithm changes.

- The reviewer is correct to point out that Hewson [1998] did indeed originally propose a contour-then-mask approach, and we thank them for pointing out this oversight. As stated in the text, the intention was to create a portable and scalable implementation of the algorithm as implemented by Berry et al. [2011b]. The underlying study by Hewson [1998] was used primarily to understand and check the mathematics of that implementation, but this particular detail of the suggested implementation, however obvious in retrospect, was overlooked. We maintain that we genuinely believed this to be an independent innovation, and not an attempt to take credit for the work of another, but have withdrawn any such claim and correctly attributed the methodology to Hewson [1998]. While this does diminish the originality of the study, we believe that highlighting and demonstrating the difference in implementation and performance still has value.

- We agree that a second order centred finite difference should be the default way to compute $\nabla^2$ on a grid, and list it in the abstract only as one of "a number of numerical improvements in the implementation", which taken together yield substantive differences compared to the previous software implementation, demonstrated in Figure 4d.

2. L159-170. Here you describe the new approach of making different resolution datasets comparable from a front perspective, and this lies at the heart of the paper's aims, particularly with regard to code provision. It is therefore of pivotal importance. You opt to adjust the smoothing whilst keeping the masking thresholds the same. I fundamentally disagree with this strategy, because, as comparison of Figs 3c and 3d shows, you can easily end up 'smoothing to death' in a way that results in the two input fields (and therefore resulting fronts) after smoothing looking almost identical, thereby destroying the whole point of having the higher resolution in the first place. A much better, much more scientifically justifiable strategy would be to do it the other way round; limit the smoothing, but adjust the thresholds, to give somewhat similar front lengths. Then any subsequent intercomparison will

show where frontal frequencies differ because of resolution impacts. In that case maybe there would need to be a bit of latitude on the amount of smoothing, perhaps a bit more for the higher resolution datasets to get rid of contaminating non-frontal noise (and there some subjectivity becomes inevitable I feel). But the very big increase from 8 smoothing passes for ERA Interim to 96 for ERA5 you use goes well beyond, to my mind, what any such latitude should allow. One might argue 96 is on a par with the number of smoothing passes (100) used in Jenkner et al (2010). Whilst one could contest that too, importantly they were using a model with a 7km horizontal resolution, much less than the 31km of ERA5. To conclude my comments here I quote from the manuscript under review: "When comparing analyses from different weather and climate datasets, the most common approach is to interpolate all the datasets to a common resolution, usually the lowest resolution among the datasets of interest. For some features such as fronts that are more easily identified in higher resolution data, this can be limiting". To me applying very heavy smoothing is just as limiting.

(a) We agree that ideally fronts would be identified at the native resolution of the dataset in question and with minimal smoothing applied. However, we found that without extensive smoothing, the amount of noise in the data was excessive and could not be addressed by careful choice of thresholds alone.

(b) The smoothed fields in Figures 3c and 3d are almost identical after smoothing, as would be expected after matching the climatological quantiles of the masking variables. However, Figure 8 demonstrates that more fronts are identified in the higher resolution dataset, even after smoothing, in line with the findings of Parfitt et al. [2017] after interpolation to lower resolution.

(c) Although the study by Jenkner et al. [2010] used a higher resolution model, it also used a different thermal variable, and a different identification method which placed fronts inside the baroclinic zone rather than on the edge, making a direct comparison difficult.

(d) Smoothing is certainly not a perfect solution, however it does have the advantage of allowing front identification to be carried out at the native resolution of the dataset being analysed, which can be useful for linking fronts with other phenomena such as precipitation, and facilitates comparison between datasets without introducing additional subjectivity through the choice of new thresholds, as we have demonstrated.

**Other Points**

1. Title: Does not seem to reflect the contents of the paper. It gives the impression, to me, that this is a case study paper, which it is not.

   *We have changed the title to "Objective identification of meteorological fronts: climatologies from ERA-Interim and ERA5"*

2. L32-33: Why is placing the front in the middle of a frontal zone, and on the warm air side of it, "very similar". Seems pretty different to me.

   *Changed from "This results in the fronts lying in the centre of a frontal zone, rather than at the leading edge as a synoptic meteorologist would typically put them. A very similar method was developed by Berry et al. [2011b], who directly applied the methods of Hewson [1998] to gridded data at $2.5° \times 2.5°$ resolution, placing fronts on the warm side of the strong temperature gradient." to "This results in the fronts lying in the centre of a frontal zone, rather than at the leading edge as a synoptic meteorologist would typically put them. Berry et al. [2011b] directly applied the methods of Hewson [1998] to gridded data at $2.5° \times 2.5°$ resolution, placing fronts on the warm side of the strong temperature gradient."*

3. L55: What is a "contemporary high resolution re-analysis". ERA5 at 31km resolution or, say, CERRA at 5.5km? Or even higher still. I would be inclined to say that means convection-resolving, which might mean of order 2km or less. So this evidently needs clarification.

   *Changed from "that is able to scale to contemporary high resolution (re-)analyses." to "that is able to scale to contemporary (re-)analyses with horizontal grid-spacings of $0.25°$ or less."*

4. L65: How do you use the u and v values at 850mb to compute front speed. If it is as per a previous publication then cite and say so as a minimum, but ideally expand here.

*This is already fully explained on Line 95, complete with citations, but cited again for completeness.".*

5. L75-77: Hewson shows this option but then highlights the limitations of this approach for curved fronts, and accordingly discards. Please take care to not imply otherwise.

*We have added the following: "In practice, most atmospheric fronts are curved and not simple one-dimensional objects. Hewson [1998] derived an alternative (their Equation 6) to Equation 1, based on the computation of "five-point mean axes", designed to mitigate the effects of frontal curvature on the evaluation of Equation 1, which can lead to noise and exaggerated frontal curvature. Although the alternative definition was preferred by Hewson [1998], we keep the definition in Equation 5 for compatibility with Berry et al. [2011b] and the numerous studies which have utilised that implementation. However, the option to use the alternative definition may be included in a future version of the code documented by this study."*

6. L75: "..as used in Berry.." – how do you know - you need to say. There is no reference in this Berry et al. paper to what method they have used. Same comment applies to lines 171-172: "used repeated applications...". Again how do you know?

*The authors have access to the original code written by Berry et al. [2011a] and subsequently applied in Catto et al. [2012] among other studies. This has been clarified in the final paragraph of Secion 1 "The code developed by Berry et al. [2011b] and applied and applied in those subsequent studies was originally developed on the European Centre for Medium-Range Weather Forecasts' (ECMWF) ERA-40 reanalysis [Uppala et al., 2005] at 1.125° × 1.125° resolution, and later applied to the ECMWF ERA-Interim [Dee et al., 2011] reanalysis at 0.75° × 0.75° resolution. However, that implementation was not easily portable due to*

*being written in a mixture of NCL and Fortran, and would not scale to the ECMWF ERA5 reanalysis at $0.25° \times 0.25°$, or other high resolution datasets."*

7. L78 & L84 & L91: "For a one-dimensional front (Type 1 front in Hewson, 1998)" would be better than "in one dimension".

   *We have changed this as suggested.*

8. L91: Not necessarily 1/root(2).

   *Expanded to: "The value of $m = 1/sqrt2$ was suggested by Hewson [1998] and we found it to be effective at the resolution of ERA-Interim (0.75°) and ERA5 (0.25°), but it may require additional tuning in very high resolution data sets."*

9. L91-93: Sentence means nothing. Please re-write from scratch.

   *Rewritten as "For a one-dimensional front, this criterion states that the magnitude of the gradient of $\theta_W$ must be greater than $K_2$ at a point a fraction of a grid length in the direction of greatest increase in the gradient of $\theta_W$, i.e. inside the adjacent baroclinic zone."*

10. L97: Why the superscripted T?

    *Standard notation for vector transposition, but only important for computation so removed to avoid confusion.*

11. L109: Degrees of what?

    *The Euclidean distance based on degrees of latitude and longitude, clarified in text. This is obviously a poor choice due to the variable physical separation between parallels with latitude.*

12. L117: "Moderately high resolution analyses such as ERA-Interim" – by today's standard this is low resolution. See point 3 also.

    *See response to next point since, same sentence.*

13. L118: "..is often narrow, frequently only one grid box wide..". I suspect this would apply across many resolutions. If you don't agree you have to provide clear evidence to justify this statement.

*Changed to "At or below the $0.75° \times 0.75°$ resolution of ERA-Interim, the region that satisfies Equation 2 is often narrow, frequently only one grid box wide."*

14. L110-125: This is very jumbled. It looks from the figure like you are using contouring-based colour-fill to mask out potential fronts that don't meet the masking criteria, but in the discussion you focus on using contouring to represent the locating diagnostic. The reader is left confused. And for a fair comparison – versus Berry et al – surely you should include the "search radius larger than one gridlength" on Fig. 1?

*Contours have been omitted from Figure 1a and references to zero contours of the Equation 2 have been removed from the description of the mask-then-join approach in order to avoid confusion, and the description expanded. The search radius is used in Figure 1 and an example is visible and now described.*

15. Figure 1b: This is nothing new. It looks basically the same as in Hewson (1998). If you think otherwise then a detailed and convincing description of why it is different needs to be added.

*We have corrected our previous oversight and correctly attributed the contour-then-mask approach to Hewson [1998]. However, we believe there is value in demonstrating the difference between the two approaches, which has now been clarified further in the text.*

16. L128: "key parameters" is vague. Need to be much more specific; presumably you mean "tuning thresholds for masking diagnostics"?

*The sentence does not refer to front identification specifically, but to objective feature identification in general, e.g., extra-tropical cyclones, therefore such a specific change would be inappropriate, also the amount*

*of smoothing is a parameter requiring tuning. The parameters requiring tuning for front identification are immediately clarified in the next sentence.*

17. L135: "local minima and maxima". This may be noise, or it may be a function of you having used the Hewson (1998) del-squared (eqn 5) approach, which is known to amplify frontal curvature, rather than the Hewson (1998) eqn 6, which does not. Needs further investigation and comment.

    *A paragraph has been added discussing this and the local front criteria applied by Jenkner et al. [2010]*

18. L139-141: Also discussed in the following reference (which is listed in the Jenkner paper you quote). So please acknowledge. - Hewson TD. 2001. Objective identification of fronts, frontal waves and potential waves. In Cost Action 78 Final Report – Improvement of Nowcasting Techniques, Lagouvardos K, Liljas E, Conway B, Sunde J (eds). European Commission EUR 19544. Cambridge University Press: Luxembourg: 285–290. At the same time a simple illustration of the 'cusp' behaviour you describe as smoothing passes increase would add quite a lot I think.

    *Citation added to the new paragraph indicated in the previous point.*

19. L144-145: There seems to me to be overall as much seasonal variation as there is latitudinal variation (whilst quantifying any difference is of course difficult given the different units, I am referring to differences across the range of values encountered for the two metrics, latitude and season).

    *This section has been rewritten and the comment made more specific*

20. L146: "relatively constant" – in time or in space?

    *In space, this section has been rewritten and the comment made more specific.*

21. L148-152: 25th and 50th percentiles seem rather arbitrary values; they also feel higher than I would have expected – giving 25% and 50% acceptance rates across the domain which is a lot. It would be nice to get a fuller picture of different percentile behaviours in some way, with also some example map plots of the diagnostics in two or more colours to show where the proposed criteria are satisfied and how they relate to the input fields. Use of percentile references is one of the genuinely newer parts of the paper and you should expand by showing more data and more related discussion.

   *Example plots and discussion have been added as an Appendix.*

22. L155-157: discussion of $K2 = 0$ in the Berry paper and the current paper is muddled and not possible for me to follow. One aspect is that (so far as I can tell) Berry et al did not use the second mask, which K2 refers to, so to say they set $K2 = 0$ is a bit misleading.

   *Clarified as follows "The second threshold $K_2$ in Equation 3 was not implemented by Berry et al. [2011b], equivalent to setting $K_2 = 0\,\mathrm{K\,m}^{-1}$ since $|\nabla \theta_W| \geq 0$ by definition."*

23. Figure 2 caption: Why zonal TFP? What does this mean? Surely you should us the full TFP here?

   *Zonal in the sense of the zonal mean, not the zonal component of TFP, since climatological percentiles are plotted by latitude and season. Removed to avoid confusion.*

24. Figure 2: Poor colour selection. Yellow is a bad colour to choose on a white background, and red and orange are a bit too similar for my liking.

   *Figure updated with thicker lines and different colour palette.*

25. Figure 3c: there are two of these.

   *Fixed.*

26. L177-178: "is valuable due to the small scale of the quantities of interest, e.g. K1=...." This statement makes no sense to me. In what way is quoting a threshold indicative of small scale?

*The thresholds used for front identification are very small order $10^{-11}$ and $10^{-6}$, therefore the stable and accurate numerical schemes are required to evaluate the fields to sufficient precision to accurately locate fronts. Clarified in text.*

27. L195-196: "...tends to relate to pre-frontal troughs...". No it doesn't. These are generally dynamically inert humidity gradients on the leading edge of the warm conveyor belt. They are discussed in Hewson (1998) and Hewson and Titley (2010). The latter paper indicates that, for operational implementation purposes, a third front mask based on theta only (so no humidity impact) is included with the aim of erasing these features. The authors might like to consider improving their study, and the resulting climatological frequencies, by doing the same.

*Thank you for the correction, changed to: "Such features were noted by Hewson [1998] and are associated with a warm conveyor belt running adjacent to the front. Hewson and Titley [2010] suggest a third masking criteria based on potential temperature rather than wet-bulb potential temperature that may be implemented in a future version of the code documented in this study."*

28. L201-202: The sentence spanning these lines does not reflect, at all, what the figure 4d shows.

*Corrected to: "Figure 4d shows that our numerical updates result in slightly lower numbers of fronts identified in across most of the northern and southern hemisphere extra-tropics, and slightly higher numbers of fronts identified in the tropics."*

29. L202: "far southern ocean" – in large part this is actually sea ice.

*Removed.*

30. Figures 4d-f and discussion thereof: It would be far more informative for the reader to see percentage change in frequency on these plots, and have that discussed instead.

   *Changed to percentage differences, also in Figures 5 and 8.*

31. Figure 4: Colour schemes are poor. One should be able to read off values without manual counting. If you are going to use colours why use monochrome red shades – the whole spectrum is available to you.

   *Colour palettes have been updated throughout to improve readability.*

32. L214: "Climatology" means "climatological quantile values".

   *Changed.*

33. Figs 6 and 7: Again a poorly chosen colour scheme. Only reds are used, which is bad as in EE above, besides which the scale selected highlights very little on panels 6a, b, c. The frequencies of cold and warm fronts would be of fundamental interest and utility, but this is all but lost due to the scales and colours selected.

   *Colour palettes have been updated throughout to improve readability. Scales were and are chosen throughout so that the highest frequency of front occurrence over the open ocean between approximately 60N and 60S (usually over the North Atlantic or Kuroshio) can be read directly from the figure legend.*

34. L219-229: Yes, interesting, but are such aspects not already in the Berry paper(s)?

   *We have added "as previously shown in Berry et al. [2011b]"*

35. L221: Gradients? Of what? Do you mean light winds?

   *Thank you, yes, we have changed to say "where winds are weaker, particularly in the horse latitudes and inter-tropical convergence zone".*

36. L227: "Somewhat surprisingly" – it would be helpful and probably illuminating in the context of this discussion to look at seasonally average SST contours and gradients (and sea ice distribution).

    *We have removed the "somewhat surprisingly", looking as SSTs contours and gradients and sea ice is beyond the scope of the current study, but we agree it would likely be illuminating.*

37. L235: It is vital to understand what you mean by aggregated. There are several possible meanings and because you don't say it's impossible for me to comment on Fig 8 or your inferences from that.

    *Clarified in text: "Aggregation is performed by counting individual fronts identified at the higher resolution passing through the lower resolution grid."*

38. L238: "Increased ability to resolve the required derivatives". I have no idea what this means.

    *Removed.*

39. L240: "where there fronts" is bad English

    *Typo, changed from "with more fronts seen where there fronts were already common" to "with more fronts seen where they were already common."*

40. L241: "a result of the increased resolution". Really? But with extreme smoothing you make the input fields look virtually the same (Fig 3c and d)? And if what you state were the case then why would an increase not be seen elsewhere?

    *Removed.*

41. L245: What do you mean by stable? Is this something to do with convection? And how does quasi stationary front frequency increase as a result? I don't understand that.

*We have edited this to say "due to the light winds associated with the ITCZ."*

42. L247-248: What does this sentence mean? The scale on Figure 10 is very different to the scale on Figure 7, which is worrying in itself, and besides which there is so much smoothing applied that for ERA5 that I don't think you could legitimately say anything about the resolution impact anyway.

    *Sentence removed, however the reduction in frequency is to be expected and is clarified in the previous paragraph in the text: "One ERA-Interim grid box contains nine ERA5 grid boxes. A perfectly straight front passing through one ERA-Interim grid box would pass through only three of the nine associated ERA5 grid boxes. Therefore, one might expect the front frequency in ERA5 at its native resolution to be approximately one third of the frequency in ERA-Interim. Comparing Figures 6 and 9 shows that this is approximately the case." The same applies to Figures 7 and 10.*

43. L250: "clearly visible". Yes, of course it will be if the scale has been adjusted in such a way as to make it more visible than on the counterpart plot (Fig. 7)! And what about other SST gradient regions – edge of Kuroshio etc?

    *Colour palettes have been updated throughout to improve readability. Scales were and are chosen throughout so that the highest frequency of front occurrence over the open ocean between approximately 60N and 60S (usually over the North Atlantic or Kuroshio) can be read directly from the figure legend.*

44. L258: "single core of a 2 year old laptop" is not a very professional or durable way to describe computational requirements.

    *Expanded to "a single core of an Intel i7-8565U based laptop with a theoretical maximum speed of 4.6 GHz"*

45. L265: This sounds like an unsubstantiated comment and should be removed or demonstrated.

*Modified to "and greater improvements are expected in lower resolution datasets for the reasons demonstrated in Figure 1*

46. L279-280: "Modest performance increases". I don't know what the basis for saying this is.

    *Clarified to: "modest increases in both the number of fronts and front points identified"*

47. L286: "No performance benefit". Likewise what is the metric used here? This needs to be introduced much earlier, and substantiated, rather than being just dropped into the conclusions from nowhere. For a broadscale picture the del-squared approach you have used is simpler and of itself is likely to be adequate in my view, because errors arising from exaggerated frontal curvature that you get with del-squared will probably not be so critical as they might be in real-time forecasting applications, but you don't discuss this at all.

    *Additional discussion has been added to Sections 3 and 3.2. The comment on the performance has been withdrawn. A more detailed comparison of the two methods would be of value since the approach preferred by Hewson [1998] has not been widely studied, but that is outside the aims of the current study.*

**References**

Gareth Berry, Christian Jakob, and Michael Reeder. Recent global trends in atmospheric fronts. *Geophysical Research Letters*, 38(21):1–6, 2011a. ISSN 00948276. doi: 10.1029/2011GL049481.

Gareth Berry, Michael J. Reeder, and Christian Jakob. A global climatology of atmospheric fronts. *Geophysical Research Letters*, 38(4):1–5, 2011b. doi: 10.1029/2010GL046451.

Jennifer L. Catto, Christian Jakob, Gareth Berry, and N. Nicholls. Relating global precipitation to atmospheric fronts. *Geophysical Research Letters*, 39(10):1–6, 2012. ISSN 00948276. doi: 10.1029/2012GL051736.

Dick P. Dee, Sakari M. Uppala, Adrian J. Simmons, Paul Berrisford, Paul Poli, Shinya Kobayashi, U. Andrae, M. A. Balmaseda, G. Balsamo, P. Bauer, P. Bechtold, A. C. M. Beljaars, L. van de Berg, J. Bidlot, N. Bormann, C. Delsol, R. Dragani, M. Fuentes, A. J. Geer, L. Haimberger, S. B. Healy, Hans Hersbach, Elías V. Hólm, Lars Isaksen, Per Kållberg, M. Köhler, M. Matricardi, A. P. McNally, B. M. Monge-Sanz, J. J. Morcrette, B. K. Park, Carole Peubey, P. de Rosnay, C. Tavolato, Jean Noël Thépaut, and Frédéric Vitart. The ERA-Interim reanalysis: Configuration and performance of the data assimilation system. *Quarterly Journal of the Royal Meteorological Society*, 137(656):553–597, 2011. doi: 10.1002/qj.828.

T. D. Hewson. Objective fronts. *Meteorological Applications*, 5(1):37–65, 1998. ISSN 13504827. doi: 10.1017/S1350482798000553.

Tim D. Hewson and Helen A. Titley. Objective identification, typing and tracking of the complete life-cycles of cyclonic features at high spatial resolution. *Meteorological Applications*, 17(3):355–381, 2010. ISSN 14698080. doi: 10.1002/met.204.

J. Jenkner, Michael Sprenger, I. Schwenk, Cornelia Schwierz, S. Dierer, and D. Leuenberger. Detection and climatology of fronts in a high-resolution model reanalysis over the Alps. *Meteorological Applications*, 17(1):1–18, 2010. ISSN 14698080. doi: 10.1002/met.142.

Rhys Parfitt, Arnaud Czaja, and Hyodae Seo. A simple diagnostic for the detection of atmospheric fronts. *Geophysical Research Letters*, 44(9):4351–4358, 2017. ISSN 19448007. doi: 10.1002/2017GL073662.

Sakari M. Uppala, P. W. Kallberg, Adrian J. Simmons, U. Andrae, V. Da Costa Bechtold, M. Fiorino, J. K. Gibson, J. Haseler, A. Hernandez, G. A. Kelly, X. Li, Kazutoshi Onogi, S. Saarinen, N. Sokka, Richard P. Allan, E. Andersson, K. Arpe, M. A. Balmaseda, A. C. M. Beljaars, L. van de Berg, J. Bidlot, N. Bormann, S. Caires, F. Chevallier, A. Dethof, M. Dragosavac, Michael Fisher, M. Fuentes, Stefan Hagemann, E. Holm, B. J. Hoskins, Lars Isaksen, P. A. E. M. Janssen, Roy Jenne, A. P. McNally, J. F. Mahfouf, J. J. Morcrette, Nick A. Rayner, R. W. Saunders,

P. Simon, Andreas Sterl, Kevin E. Trenberth, A. Untch, D. Vasiljevic, P. Viterbo, and J. Woollen. The ERA-40 re-analysis. *Quarterly Journal of the Royal Meteorological Society*, 131(612):2961–3012, 2005. doi: 10.1256/qj.04.176.

---

## Author Comment (AC3)

**Response to Jatin Katla for GMD-2022-255**

Philip G. Sansom and Jennifer L. Catto

December 2023

In reviewing the manuscript, I found a result which is very intriguing. The authors have acknowledge this result, but not really explained or carried out further analysis, that i think is warranted. I refer to this result:

"Somewhat surprisingly, cold fronts are slightly more common though less widely distributed in the Southern Hemisphere during southern summer (DJF, Figure 7(a)) than in southern winter (JJA, Figure 7(c))."

This is indeed very surprising, and runs counter-intuitive. When I focus on the region of southwest Western Australia (where I live and regularly check MSLP charts), the analysis shows a higher frequency of cold fronts in DJF as compared to JJA. I find this very odd, and would like the authors to dig a little further.

Regions with Mediterranean climates, such as southwest WA, get most rainfall in Winter (JJA), and the heaviest rain events, are most commonly associated with cold fronts. Yet, Figure 7 suggests there are more cold fronts in Summer than Winter, which is very counter-intuitive. It is generally accepted that cold-fronts bring rain, and it would not be un-reasonable to assume, at least based on first principle, that where you have more frequent cold fronts, one might expect more rainfall. Your results suggests the opposite.

*There are indeed more cold fronts apparent over south-western Western Australia (WA) itself in southern Summer (DJF). A similar phenomenon is visible in the original climatologies of Berry et al. [2011b] and Berry et al. [2011a]*

*who noted that these are non-precipitating fronts associated with strong moisture gradients between the ocean and adjacent hot dry areas a short distance inland. There are also many cases of non-precipitating fronts that contribute to severe fire weather (e.g., https://www.nature.com/articles/s41612-023-00425-z). The revised colour scheme in Figure 7 shows that over the frequency of cold fronts over the ocean to the west of south-western Western Australia in south Winter (JJA) does increase as expected, with a coherent area of heightened cold front frequency (compared to the surrounding area) extending over the Mediterranean portion of south-western Western Australia.*

**References**

Gareth Berry, Christian Jakob, and Michael Reeder. Recent global trends in atmospheric fronts. *Geophysical Research Letters*, 38(21):1–6, 2011a. ISSN 00948276. doi: 10.1029/2011GL049481.

Gareth Berry, Michael J. Reeder, and Christian Jakob. A global climatology of atmospheric fronts. *Geophysical Research Letters*, 38(4):1–5, 2011b. doi: 10.1029/2010GL046451.

---

## Referee Report (RR1)

- Title/Abstract: The new title puts the focus on the climatologies when they are more a by-product of the improved methodology. The abstract especially is missing any climatological results. Please remove this discrepancy by adapting the title (replacing the colon by "and" would already go a long way) and Abstract (add a couple sentences on the climatological results, if necessary at the expense of some detail on the methodological adaptations).
- Figures: Add "UTC" to datetimes in the captions.
- Figure 2: The legend to me implies that each line represents a season (e.g., DJF in blue), so I first thought three years were shown, when in fact each line corresponds to one month but the three months of each season are colored the same. Listing each month separately in the legend would resolve that. Also, the horizontal dotted lines are almost invisible (printout); change that, and list the latitude values in the caption.
- Figures 7/10: Consider shortening the caption to "... (a--d) cold fronts and in (a,e) DJF, (b,f) MAM, ...". Either way, don't unnecessarily capitalize "cold fronts" and "warm fronts".
- Lines 5/6: "Smoother fronts with fewer breaks" doesn't necessarily sound like a drawback of the original method. Please reformulate to express why "distorted fronts with many breaks" are actually more desirable (as implied by the sentence).
- Line 15: Add references for "modelling" as well as more than one for the "numerous" case studies.
- Line 78: Consider introducing an acronym for "Hewson (1998)", e.g., "H98", given how often it's referenced in the text.
- Line 103: Use "ABZ" or don't define it in the first place.
- Line 109: Use "K3" in Equation 4 (like K2 in Equation 3).
- Line 115: The descriptions of the two approaches are a bit hard to follow. Refer to Figure 1 at the beginning of each description so the reader is aware of this visual aid while reading the test. Also, consider naming the equations when referencing them (e.g., (1) TFL, (2) TFP, (3) "ABZ", (4) "front speed"; "... that satisfy te TFP Equation 2 to form a mask (the ABZ criterion in Equation 3 is ...") so the reader is spared from memorizing the equation numbers or jumping back and forth in the text.
- Line 156: The presented method uses "three parameters", but that's not necessarily true of "front identification" methods in general.
- Line 199: Capitalize Northern/Southern Hemisphere (here and elsewhere).
- Line 206: Add references for "previous studies".
- Line 273: To what degree is the "increase by almost 100%" due to larger vs. newly identified fronts? Add an estimation if you can provide one, or at least mention that both effects play a role (assuming that's the case).
- Lines 288/289: I had to look up "horse latitudes". Consider a more common term (unless you deem this common knowledge). Also, define "ITCZ" at first use (currently the definition is on line 309).
- Lines 294/295: Please discuss why cold fronts are more common in SH summer than winter when the opposite is true in the NH (while frontal precipitation nevertheless peaks in SH winter; see editor's remark and your response to is).
- Line 308: It is unclear whether the increase is in the range of 20--40%, or whether the increase is by 20% from 20% to 40%. Please reformulate.
- Discussion: Please revise the text, as it's in a markedly rawer state textually than the rest of the paper.
- Line 328: Given NCL is mentioned for the old implementation, also mention R for the new one.

- Lines 342/343: Either remove the sentence "Computational performance ...", or move it to the earlier paragraph where the performance improvements are discussed.

---

## Author Response (AR2)

**Response to Second round of reviews GMD-2022-255**

Philip G. Sansom and Jennifer L. Catto

April 2024

**Reviewer 1**

- Title/Abstract: The new title puts the focus on the climatologies when they are more a by-product of the improved methodology. The abstract especially is missing any climatological results. Please remove this discrepancy by adapting the title (replacing the colon by "and" would already go a long way) and Abstract (add a couple sentences on the climatological results, if necessary at the expense of some detail on the methodological adaptations).

*Title has been changed as suggested.*

- Figures: Add "UTC" to datetimes in the captions.

*This has been added as suggested.*

- Figure 2: The legend to me implies that each line represents a season (e.g., DJF in blue), so I first thought three years were shown, when in fact each line corresponds to one month but the three months of each season are colored the same. Listing each month separately in the legend would resolve that. Also, the horizontal dotted lines are almost invisible (printout); change that, and list the latitude values in the caption.

*We have changed the caption to make this clearer. The horizontal dashed*

*lines appear very clearly when I print out the page, so we have kept the lines as they are, so as not to add extra clutter to the figure.*

- Figures 7/10: Consider shortening the caption to "...  (a–d) cold fronts and in (a,e) DJF, (b,f) MAM, ...". Either way, don't unnecessarily capitalize "cold fronts" and "warm fronts".

*We have edited the caption as suggested, and have not added any unnecessary capitalisation.*

- Lines 5/6: "Smoother fronts with fewer breaks" doesn't necessarily sound like a drawback of the original method. Please reformulate to express why "distorted fronts with many breaks" are actually more desirable (as implied by the sentence).

*The reviewer is correct that we believe smoother fronts with fewer breaks is desirable. The "original method" referred to here is the Hewson method of contouring. We have edited the text to make this clearer.*

- Line 15: Add references for "modelling" as well as more than one for the "numerous" case studies.

*Added additional references and dropped the "numerous"*

- Line 78: Consider introducing an acronym for "Hewson (1998)", e.g., "H98", given how often it's referenced in the text.

*We have done this as suggested.*

- Line 103: Use "ABZ" or don't define it in the first place.

*Changed as suggested.*

- Line 109: Use "K3" in Equation 4 (like K2 in Equation 3).

*Changed as suggested.*

- Line 115: The descriptions of the two approaches are a bit hard to follow. Refer to Figure 1 at the beginning of each description so the reader is aware of this visual aid while reading the test. Also, consider naming the equations

when referencing them (e.g., (1) TFL, (2) TFP, (3) "ABZ", (4) "front speed"; "... that satisfy te TFP Equation 2 to form a mask (the ABZ criterion in Equation 3 is ...") so the reader is spared from memorizing the equation numbers or jumping back and forth in the text.

*We thank the reviewer for this suggestion to make it easier to read the text. We have added these names to the description as suggested.*

- Line 156: The presented method uses "three parameters", but that's not necessarily true of "front identification" methods in general.

*This has been corrected.*

- Line 199: Capitalize Northern/Southern Hemisphere (here and elsewhere).

*This has been corrected.*

- Line 206: Add references for "previous studies".

*Added several additional references.*

- Line 273: To what degree is the "increase by almost 100%" due to larger vs. newly identified fronts? Add an estimation if you can provide one, or at least mention that both effects play a role (assuming that's the case).

*This is difficult to quantify, so we have mentioned both effects as suggested.*

- Lines 288/289: I had to look up "horse latitudes". Consider a more common term (unless you deem this common knowledge). Also, define "ITCZ" at first use (currently the definition is on line 309).

*ITCZ is now defined earlier. The latitudes are now given along with the phrase "horse latitudes".*

- Lines 294/295: Please discuss why cold fronts are more common in SH summer than winter when the opposite is true in the NH (while frontal precipitation nevertheless peaks in SH winter; see editor's remark and your response to is).

*We have added a reference to Berry et al. [2011] and Satyamurty and de Mat-*

*tos [1989]*

- Line 308: It is unclear whether the increase is in the range of 20–40%, or whether the increase is by 20% from 20% to 40%. Please reformulate.

*This has been changed to read "where front frequency increases by between 20 % and 40 %."*

- Discussion: Please revise the text, as it's in a markedly rawer state textually than the rest of the paper.

*It is difficult to know exactly what is required here. We have tried to make the text more concise.*

- Line 328: Given NCL is mentioned for the old implementation, also mention R for the new one.

*This has been added.*

- Lines 342/343: Either remove the sentence "Computational performance ...", or move it to the earlier paragraph where the performance improvements are discussed.

*This has been removed as suggested.*

**Reviewer 2**

This is the second review I have provided for this manuscript. As such I can first say that it is good to see that the record has been put straight regarding correct attribution of methodologies for creating objective front plots, following on from extensive comments in the first review round. Another big plus for this revised paper is the supplementary material, which supports the main scientific (methodological) achievement of the paper, namely the use of the climatological quantiles to define the masking criteria; accordingly more pointers should in my view be made to this within the main text.

*We thank the reviewer for their positive assessment of the revised manuscript.*

*We have made more reference to the supplementary material as suggested.*

My primary comment now concerns the description of the motivation and objectives of the paper. In terms of scientific/meteorological content the paper does not go far into describing or explaining the features on show from the different re-analyses – e.g. Figure 10 attracts but one short paragraph of somewhat sketchy comments. Indeed such content, overall, falls short of what would be needed for acceptance in a "standard" meteorological journal (e.g. Monthly Weather Review). On the other hand, the coding and threshold setting aspects, and thereby the provision of a tool for the community to use do fit better the aims of GMD, which can justify publication there, and so the authors should be clearer on that aspect from the outset, stating this more clearly as the purpose of the paper (in the abstract and, within the introduction, at the start and not just at the end).

*We have stated the main aim of the paper early in the abstract as suggested.*

Following on I really would like the authors to rewrite the abstract. It is particularly jumbled at the moment, and will in my view put readers off! This jumbled nature is a legacy of poor attribution of previous work in the first submission, and incomplete attempts to address that in the wording this time. For example: both "implemented a number of changes to a previous implementation" and then "previous implementation used a different order compared to the original algorithm" are confusing, especially taken together. You might like to say something like "we have resurrected the original methodology, used this as the basis for a new and modern open source code implementation, and show how this delivers output which in various ways is rather better than a previous code implementation which was unwieldy, slow to run, and deviated too much in its methodological approach from the original work". By all means then go on to reference the additional changes you have made, but I stand by my comment from the first review that "more accurate finite differencing" is not really your work. Neither is it innovative, or new, and so it should not appear in the abstract as if it were.

*We thank the reviewer for the comments and have now edited the abstract to (hopefully) be clearer.*

Related also to the additional changes, Section 3.4, on numerical updates, is

problematic for various reasons and needs a bit of a re-write (details below). The remaining comments I provide below should generally be fairly easy to address, but are nonetheless important for final acceptance, in my view.

*We have responded individually to the comments below.*

Other points 1. L1 - "are important for their..." is poor English; please reword.

*Changed.*

2. L14 - what is "large proportion of total and extreme precipitation"? This is poor English.

*Added the word "both".*

3. L24 - in the view of most I think 1998 would not be classified as a recent year.

*Have removed "in recent years".*

4. L34 - not sure this can be "the final piece" of the puzzle as you go on to criticise this.

*Changed.*

5. L44 – you cannot really introduce "the threshold" in this way without saying what it is. Readers will be confused.

*This sentence has been removed.*

6. L48 – "shown lead" – English error.

*Corrected.*

7. L57-58 – "The aim" ... "as implemented by Berry et al" is not really the aim here. You are actually ditching the Berry et al implementation in many ways. Please correct. This is a really important point as indicated in "summary / main points" above.

*We have changed these lines.*

8. L90 – "of the equation 1"; delete "the"

*This has been corrected.*

9. L104-5 – it may require additional tuning in very high resolution datasets. This is an interesting comment which I don't think I agree with. Anyway to justify inclusion you would in my view need to say more about why you think this is the case. Otherwise leave this out.

*This has been removed as suggested.*

10. L109-111 – you need to say why you use the -1.5 to +1.5m/s range for quasi-stationary.

*It is stated at the start of the sentence that we have followed Berry et al.*

11. L155 – "usually necessary" – surely it's always necessary to define what you are doing!

*We have added the word "subjectively".*

12. L155-156 – "it is still necessary to provide certain configuration settings" would be better here; using the term "parameter" here is a bit confusing.

*See previous response.*

13. L164 –To qualify your statement that you particularly want to avoid "unwanted" local extrema in the TFL you do need to say why you expect them. You deal with this in the next paragraph, so maybe some text rearrangement here would be the right approach.

*The text has been rearranged as suggested.*

14. L170 – It think you can replace "may in part be" with "will in part be".

*Changed as suggested.*

15. L171-2 – designed to "quell the amplification of frontal curvature" would

be a more accurate reflection of the contents of that paper.

*Text has been changed as suggested.*

Also, I really would strongly recommend that you reference the frontal "hook" seen south of Iceland on Figure 3d as a very nice example of the detrimental impact of not using equation (6). This looks (and would be) important locally, for forecasting applications, but is very probably not for your 'climatological' purposes, which gives to my mind a nice visual justification for your simplified approach.

*We appreciate the reviewer's suggestion regarding the hook seen in the frontal structure south of Iceland. We have thought very carefully about this. Looking at Figure 8 in Hewson [1998], there are some similar hook type structures identified even when using Equation 6 from that paper. As the more stringent masking criteria are applied, these hook features are removed. We don't believe it is possible to say for sure that the hook feature seen in our Figure 3 would not be present if we had used the Hewson [1998] Equation 6.*

16. L159 – practically, such charts will never all be produced by a "single meteorologist", but rather a set of meteorologists working shifts (which of course all bring their own, sometimes different, subjective judgements to the table).

*The text has been changed slightly to reflect this.*

17. L158-9 and L187 – Is there a bit of a contradiction here; first you criticise the cross-referencing to charts, yet then go on to do that yourselves?

*We have added a sentence of qualification "While comparing to charts is a necessary check of an objective algorithm, calibrating in this way..."*

18. L182 – "a threshold of..." – for what parameter? K1 or K2?

*$K_1$ has been added here.*

19. Fig. 2 caption: should say what period, in years, this data is for, and what the input dataset what, at what resolution and with what smoothing.

*The time period and input dataset and smoothing level have been added.*

20. L227-228 – you used January 2000, then say that was consistent with Figs 1 and 3, but those figures are for 2001.

*We thank the reviewer for pointing out this typo, January 2001 was used.*

21. L236-238 – the explanation of the del-squared computation is incomplete / confusing / incorrect. Surely it's one gridbox either side? There are no "components" as such to del-squared.

*We thank the reviewer for helping us clarify this area of the text. Yes it is only one grid box, that sentence was a hangover from when fourth order accurate derivatives were considered, and has been removed. And, yes strictly there are meridional and zonal terms, not components, to del-squared and this had been clarified in the text.*

22. L239 – what are the edges of the domain? I didn't think a global domain had any edges?

*In Cartesian coordinates, the east/west edges of the domain can be handled by wrapping, but the north/south edges cannot, and if it is not necessary to run the analysis globally, a smaller area may be considered. We believe the readers will understand this.*

23. L239-242 – the fact that thresholds are supposedly "very small" does not in any way mean that we need accurate and stable schemes. Small is merely a function of the unit. If we were working in microns they would not be nearly "as small". So this sentence does not make sense and I really don't know what you are trying to say here. Please re-think.

*This sentence has been removed and the computational efficiency clarified.*

24. L243-245 – This sentence does not make sense. I am not quite sure what you aim to say here, but it would at least be best to start by referring to the "standard relative humidity parameter provided by ECMWF which accounts for ....". Then go on to say (I think!) what the NCL code requires as input.

*The intention was simply to be transparent about a change in methods. The text has been updated to clarify this point.*

25. L245 – How did Berry in 2011 use a tool from 2019?

*The reference for NCL has been backdated and moved to where NCL is first mentioned.*

26. L250 – why250km?

*This is a subjective choice and a sentence of explanation has been added.*

27. L259-260 – there are two warm fronts here. Please clarify for reader's which one you are referring to (it's evidently the southern one).

*Clarified in text.*

28. L261 – "suggest..." - I think "use..." would be a more accurate reflection of this paper's contents.

*This has been changed.*

29. L271 – the "greatest increases". This is incomplete / misleading / wrong. First you need to define what you mean by increases; reference to figures 4e and 4f suggests that you mean in percentage terms, yet the areas you highlight are more like minima, notably on Fig 4e. Then if you do mean in absolute numbers we can't see that clearly. I think this issue may be a legacy of the figure having been changed to a percentage change, following the previous review round, but the text not made compatible. So this clearly needs fixing.

*Text has been updated to reflect the shift to percentage changes*

30. L272 – whatever way you look at it, I don't think this can be said to be evidence of "the effectiveness of the contour-then-mask approach..". We have already seen evidence of that aspect, so that should in my view be enough and I would leave this statement out.

*We have removed this statement as suggested.*

31. L273 – by almost 100%. This is not correct. My inspection gives about 40% (from about 6 to about 8.5).

*Thank you for pointing this out. We have changed this to say between 40 and 80%, and referenced the figure panel where this is seen.*

32. Figure 4 caption line 1: English - please change to "...front frequency in % (using 6-hourly frames from ....)." reminding users what period is under investigation from what re-analysis after the "from".

*This has been edited as suggested.*

33. Figure 4e, f, caption, Figure 5 caption and other locations – whenever you refer to percentage difference/change it's vital to say how that is computed, as various approaches are possible. For Fig. 4e for example is it: (100*(c-a)/a) or (100*(c-a)/c) or (100*(c-a)/(2*(a+c))) or some other variant. And then one could also ask what happens when the denominator is nil.

*We have added this information. There are no grid boxes where the denominator is zero.*

34. L275: maybe you should refer to Figure 6 here first, before discussing Figure 5 directly, as it provides input?

*Figure 6 has now been referenced here.*

35. L280: edges would be better than edge.

*Edited as suggested.*

36. L294-295. It would be nice to see some discussion of possible reasons why there is a difference between the SH and NH - i.e. in the NH we clearly have more cold fronts in winter than in summer.

*We have added a reference to Berry et al. [2011] and also to Satyamurty and de Mattos [1989] regarding the frontogenesis locations.*

37. L295-296. I am not sure what you are trying to say here – it's ambiguous. Do you mean to imply that on average the storm track moves poleward in SH winter. If so I don't agree that that is the case. One can see for example in Hoskins and Hodges (2005) that the storm track zone expands equatorward in the SH winter. Or maybe you mean that there are specific occasions that the storm track strays poleward in SH winter that lead to a higher warm front

density near Antarctica than in SH summer. I don't think that that would be true either though, and I think the cause may actually have something to do with sea ice and/or katabatic drainage, but I am not sure exactly what. Anyway, please clarify or remove.

*We have edited this sentence to remove the implication that the storm track moves polewards.*

38. L306. "...more fronts are identified...". Any thoughts on why this is the case?

*This could be related to the way the aggregation is done. The higher resolution dataset could have two individual fronts identified that pass through the same larger grid box. In the lower resolution dataset this would count only as a single frontal point. We have added a line of explanation: "Since aggregation is performed by counting individual fronts, this indicates that ERA5 is able to resolve more fronts due to its higher resolution".*

39. L307. Any idea why there might be a reduction over high orography?

*This is likely related to the better representation of orography in the higher resolution dataset.*

40. L308. "20% to 40%" must be changed to "20 to 40%" otherwise the meaning is wrong.

*This text has been changed.*

41. L308-310. So what are the ITCZ fronts? The ITCZ is not a place where one generally expects to see fronts, I think, or am I missing something?

*These could be similar to the fronts identified in the South Pacific Convergence Zone, with strong temperature gradients. We have added that the numbers are still very small so this large percentage increase is still a very small absolute increase in the frequency. We have also referenced Figure 9d here instead of later.*

42. L313-314. "would typically pass through" is the correct terminology.

*This has been edited as suggested.*

43. L315. It would be much easier to see this if you made the scale divisions on Fig 9 equal to one- third of the size of the divisions on Fig 6, instead of one quarter! Please think of the reader!

*We prefer not to have the scale divisions at 1/3 etc. It is only an approximation, and therefore we have chosen to keep the scale as it is.*

44. L321. SST fronts would be fronts in the ocean. Therefore please clarify that you are referring to atmospheric fronts which are to some extent indicative of underlying or nearby SST structures, with the connecting physical mechanism being vertical fluxes of heat and moisture.

*We have now mentioned that the high frequency of atmospheric fronts associated with the SST fronts are visible.*

45. Figure 8a, b and Figure 9a. Whilst the colour schemes adopted on the figures are overall a big improvement on the previous manuscript version, in these panels we have descended into some sort of "multicolour mayhem" making visual interpretation difficult. I see you have a scheme from ColorBrewer (I found it here: https://en.wikipedia.org/wiki/ColorBrewer). I think this particular scheme would be recommended for use with multiple paired classes, not for a data continuum that you have. One particular issue is that red draws the eye, and here it represents mid-range values. Unless there is a particular physical reason to use a particular colour in a particular part of the range (which is not really the case here) then brighter colours like red should usually be reserved for more extreme values. There are a couple of recorded presentations here on colour usage that might be of interest: https://vimeo.com/717994549/76f08433b8 and https://vimeo.com/718387621/4ec07e604b.

*We thank the reviewer for the helpful comments and the links to the presentations. Since the reviewer believes the figures are improved from the last version, and so as not to have to reproduce every figure, we have elected to keep the colour scheme as it is.*

46. L 342-343 – This sentence duplicates what was said a little earlier on and can be deleted.

*This has been removed.*

47. L347 – deleting "of the masking variables" would make for a sentence with better English.

*We thank the reviewer for this suggestion, but choose to keep in "of the masking variables" so that it is clear where these variables are used.*

48. L358-361 – I don't think that moving to higher order accuracy in the finite differencing in this way is really something to be countenanced here. Such schemes have their roots, I believe, in improving the fidelity and accuracy of numerical scheme implementations, where accurate forward integration is of paramount importance, whereas here the goal is replicating at a local scale the instantaneous model-derived picture. There is no forward integration. The higher order schemes inevitably bring into play more data from points remote from the local area, so can contaminate the local picture that the user wants to focus on. Indeed this was in a sense something that you criticised the Berry et al code for earlier in the manuscript. So I would strongly recommend that you either discuss this aspect in full, or just leave this part of the text out. Just having a modest increase in the number of front points identified is, of itself, neither good or bad, so I really don't think there is much useful to say here.

*We have removed these points.*

49. L364-365 – see bullet point 15 above.

*Text has been edited in line with previous comment.*

50. I had a quick look at the Zenodo link. Please remember to update the publication link there when ready.

*We will ensure we change the link when we are ready.*

51. Appendix L8. "of the time"?? =manually produced charts for that particular time? If so which ones, from which centre (there can be a lot of differences between those too, depending on country of origin, which may incidentally also be a point worth making earlier in the manuscript).

*We have changed this to "that may be identified by a synoptic meteorologist may not be correctly identified".*

52. Appendix L18. Ditto.

*We now refer to Met Office charts from that date.*

53. Appendix. Perhaps on Figures 2, 3, 4 you could put a box around the plots to show which thresholds you use in the end, to help the reader, or add a big star, or something equivalent?

*We thank the reviewer for this suggestion and have highlighted in the caption which threshold is used.*

**References**

Gareth Berry, Michael J. Reeder, and Christian Jakob. A global climatology of atmospheric fronts. *Geophysical Research Letters*, 38(4):1–5, 2011. doi: 10.1029/2010GL046451.

T. D. Hewson. Objective fronts. *Meteorological Applications*, 5(1):37–65, 1998. ISSN 13504827. doi: 10.1017/S1350482798000553.

Prakki Satyamurty and Luiz Fernando de Mattos. Climatological Lower Tropospheric Frontogenesis in the Midlatitudes Due to Horizontal Deformation and Divergence. *Monthly Weather Review*, 117(6):1355–1364, 1989. doi: 10.1175/1520-0493(1989)117¡1355:CLTFIT¿2.0.CO;2.